# CLIP Meets Diffusion: A Synergistic Approach to Anomaly Detection

**Byeongchan Lee**                                                      *prinsommer@kaist.ac.kr*
*Korea Advanced Institute of Science and Technology (KAIST)*

**John Won**                                                              *johnwon@kaist.ac.kr*
*Korea Advanced Institute of Science and Technology (KAIST)*

**Seunghyun Lee**                                                       *shyun4839@kaist.ac.kr*
*Korea Advanced Institute of Science and Technology (KAIST)*

**Jinwoo Shin**                                                           *jinwoos@kaist.ac.kr*
*Korea Advanced Institute of Science and Technology (KAIST)*

**Reviewed on OpenReview:** *https://openreview.net/forum?id=WpFzZNuQmg*

## Abstract

Anomaly detection is a complex problem due to the ambiguity in defining anomalies, the diversity of anomaly types (e.g., local and global defect), and the scarcity of training data. As such, it necessitates a comprehensive model capable of capturing both low-level and high-level features, even with limited data. To address this, we propose CLIPFUSION, a method that leverages both discriminative and generative foundation models. Given the CLIP-based discriminative model's limited capacity to capture fine-grained local details, we incorporate a diffusion-based generative model to complement its features. This integration yields a synergistic solution for anomaly detection. To this end, we propose using diffusion models as feature extractors for anomaly detection, and introduce carefully designed strategies to extract informative cross-attention and feature maps. Experimental results on benchmark datasets (MVTec-AD, VisA) demonstrate that CLIPFUSION consistently outperforms baseline methods in both anomaly segmentation and classification under both zero-shot and few-shot settings. We believe that our method underscores the effectiveness of multi-modal and multi-model fusion in tackling the multifaceted challenges of anomaly detection, providing a scalable solution for real-world applications.

## 1 Introduction

Anomaly detection is an important problem in real-world applications such as industry (Bergmann et al., 2019; Zou et al., 2022) and medicine (Setio et al., 2017; Menze et al., 2014). In this paper, we address anomaly classification and segmentation.[1] In anomaly classification (or segmentation), each image (or pixel) is classified as normal or anomalous.

In practice, the number of abnormal images is small, while the potential types of abnormalities are highly diverse, making it challenging to obtain representative examples. Moreover, developing methods that can quickly adapt to new environments (e.g., new processes or products), even with a limited number of normal images, has become a critical research topic (Rudolph et al., 2021; Sheynin et al., 2021; Zou et al., 2022; Huang et al., 2022). To address this, we focus on the problem within the zero-shot or few-normal-shot[2] setup.

---

[1]It is also referred to as anomaly detection and localization, respectively, in the literature. We use the terms anomaly classification and segmentation, and refer to anomaly detection as the overarching problem encompassing both.

[2]In this paper, "few-shot" specifically refers to "few-normal-shot".

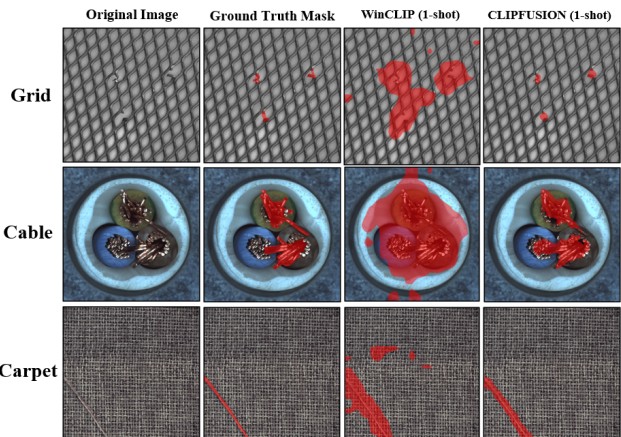

Figure 1: Qualitative results of our model CLIPFUSION. It captures fine-grained anomalies more effectively than WinCLIP.

The recent advent of large-scale pretrained foundation models, e.g., GPT (Brown et al., 2020) and DALL-E (Ramesh et al., 2021; 2022), has opened new horizons for addressing such challenges. The generalizability and scalability of foundation models make them particularly effective for tackling zero-shot and few-shot tasks such as anomaly detection. Effectively utilizing off-the-shelf foundation models is an active research direction. Our design follows this trend without retraining or introducing a shared module. Building on this, we present a novel methodology that combines the complementary strengths of discriminative and generative foundation models.

Anomaly detection is a multifaceted problem, and various approaches have been developed based on different perspectives. First, anomaly detection has the aspect of one-class classification. This has led the community to methods that learn normality from normal images and judge what deviates from it as anomalous. Here, many methods extract normal features from normal images using a vision foundation model (Cohen & Hoshen, 2020; Defard et al., 2021; Roth et al., 2022). Second, anomaly detection has the aspect of two-class classification. The community has employed vision-language foundation models, such as CLIP (Radford et al., 2021), to compare a query image against normal and abnormal text prompts (Jeong et al., 2023; Li et al., 2024a). Third, anomaly detection also has the aspect of semantic segmentation. Many anomalous features are fine-grained ones that appear in textures or edges. WinCLIP (window-based CLIP) attempted to solve this by sliding windows at multiple scales. However, it is known that CLIP has limitations in learning fine-grained features (Wysoczańska et al., 2025; Jiang et al., 2023), and such an approach might be suboptimal. Also, it requires hundreds of forward passes due to the exhaustive sliding of multi-scale windows.

The multifaceted nature of anomaly detection makes it challenging to address using only discriminative models. So, we aim to create synergy by fusing a discriminative vision-language model (VLM) with a generative text-to-image (T2I) model. These are two fundamentally distinct approaches to aligning images and text. Discriminative models, such as CLIP, directly align embeddings in the same space, while generative models, like diffusion models, align them indirectly by jointly performing a common task. Diffusion models align images and text through a generative process that preserves fine-grained semantics. Consequently, they may capture more detailed visual patterns. Therefore, it can be beneficial to leverage both high-level information extracted from the CLIP-based model and low-level information extracted from the diffusion-based model for effective anomaly detection.

We propose CLIPFUSION (= CLIP + Diffusion), a novel method that combines the strengths of multi-modal (language and vision) and multi-model (CLIP and diffusion) approaches. In the zero-shot scenario, CLIPFUSION effectively identifies anomalies by integrating image-text alignment from a CLIP-based model and cross-attention maps (Chefer et al., 2023; Zhang et al., 2023b; Liu et al., 2024) from a diffusion-based model. In the few-shot scenario, it further enhances its performance by additionally leveraging feature

maps from the CLIP image encoder and the diffusion denoiser, enabling a more comprehensive comparison between the query image and reference images. Figure 1 illustrates qualitative results, showcasing the model's ability to capture fine-grained anomalies. CLIPFUSION exemplifies a holistic framework that synergizes complementary foundation models, and has the potential to extend beyond anomaly detection and advance other recognition tasks.

**Contributions** of our work are summarized as follows:

- We propose the use of diffusion models as feature extractors for anomaly detection. To this end, we exploit cross-attention and feature maps and carefully design extraction strategies. Since we extract intermediate features from an off-the-shelf diffusion model without any training or iterative denoising, our method is fast and efficient.

- We introduce a novel approach to anomaly detection that jointly leverages features from CLIP and diffusion models. We demonstrate the complementary benefits of combining features from both models. Since diffusion models compensate for CLIP's limitations in capturing fine-grained local details, our method eliminates the need for time-consuming techniques such as multi-scale sliding windows.

- In terms of performance, our method consistently outperforms baselines. Unlike most prior works, it is also versatile in scope: it can handle both segmentation and classification tasks under zero-shot and few-shot settings.

- Our method requires no additional training, which enables fast real-world deployment. Even in few-shot settings, the samples are used solely for feature extraction rather than training, making the method more robust. In contrast, training-based methods are sensitive to the choice of when to stop training, which is challenging in few-shot setups due to the absence of a validation set.

## 2 Related Work

**Vision(-Language) Models for Anomaly Detection** In works that utilize vision(-language) models in anomaly detection, features are extracted from the intermediate layers of a model pretrained on a discriminative task (e.g., classification or contrastive learning). Anomaly detection is performed by comparing the features of the query image with the features of reference images. Some works utilize ResNet-based models pretrained on ImageNet in the few-shot regime. In SPADE (Cohen & Hoshen, 2020), a $k$-nearest neighbors algorithm is employed to identify the $k$ reference images closest to the query image, which are then used as the basis for anomaly detection. In PaDiM (Defard et al., 2021), features extracted from reference images are modeled as Gaussian distributions, and the features of the query image are compared against these distributions. In PatchCore (Roth et al., 2022), a memory bank is constructed using the features of reference images, and the features of a query image are directly compared against this memory bank. WinCLIP (Jeong et al., 2023) applies local windows and combines the results through harmonic aggregation. PromptAD (Li et al., 2024a) learns prompt tokens through contrastive training.

A separate line of work assumes access to additional data and leverages it. Some methods require an auxiliary dataset containing both normal and anomaly images, e.g., training on the MVTec-AD dataset and inference on the VisA dataset (Hu et al., 2024; Zhu & Pang, 2024; Yao et al., 2024; Li et al., 2024c; Cao et al., 2024; Zhou et al., 2023; Gu et al., 2024a; Qu et al., 2024). AnomalyGPT (Gu et al., 2024b) relies on an auxiliary dataset with a large number of normal samples. In practice, acquiring such data with matching characteristics is challenging, and these methods often struggle when deployed on datasets with distribution shifts. Moreover, the need for additional training increases deployment time. Some methods also rely on ground-truth segmentation masks, which are difficult to obtain in real-world scenarios (Hu et al., 2024; Cao et al., 2024; Zhou et al., 2023). MuSc (Li et al., 2024b) leverages the test set by comparing the features of test images, and is only feasible for batch-processing of the test set.

**Diffusion Models for Discriminative Tasks** Recently, there has been growing interest in applying diffusion models to discriminative tasks (Fuest et al., 2024). In DDPM-Seg (Baranchuk et al., 2021), feature

maps are constructed using activations extracted from intermediate layers of the denoiser and are utilized for semantic segmentation. VPD (Zhao et al., 2023) further incorporates cross-attention maps for semantic segmentation and depth estimation. These methods require fine-tuning for downstream tasks. On the other hand, DiffusionClassifier (Li et al., 2023a) performs zero-shot classification by determining which text conditioning best predicts the noise. P2PCAC (Hertz et al., 2022) and USCSD (Hedlin et al., 2024) reveal that cross-attention maps contain rich information about spatial layouts through image editing and semantic correspondence tasks.

## 3 Preliminaries

**CLIP**   CLIP (Radford et al., 2021) consists of an image encoder $f_I(x)$ and a text encoder $f_T(c)$. These encoders map images and texts into the same embedding space. The model is trained to match correct image-text pairs with high similarity while assigning low similarity to incorrect pairs. This process is optimized using the following contrastive loss (Chen et al., 2020):

$$p(T = \hat{c}|I = x) := \frac{\exp(\text{sim}(f_I(x), f_T(\hat{c}))/\tau)}{\sum_{c \in \mathcal{C}} \exp(\text{sim}(f_I(x), f_T(c))/\tau)}, \tag{1}$$

where $\mathcal{C}$ is the set of candidate text prompts, $\tau > 0$ is the temperature hyperparameter, and $\text{sim}(\cdot, \cdot)$ is the cosine similarity measure.

**Diffusion**   A diffusion model (Ho et al., 2020) is trained through a sequential process that involves forward diffusion and backward denoising. In the diffusion process, random noise is progressively added to a clean image over several steps, following the equation:

$$q(x_t|x_{t-1}) = \mathcal{N}(x_t; \sqrt{1 - \beta_t}\, x_{t-1}, \beta_t\, \mathbf{I}), \tag{2}$$

where $\beta_t$ is the noise schedule hyperparameter, which determines the amount of noise added to the image $x_{t-1}$ (Nichol & Dhariwal, 2021). In the denoising process, random noise is progressively removed over several steps using the following equation to restore the original image:

$$p_\theta(x_{t-1}|x_t) = \mathcal{N}(x_{t-1}; \mu_\theta(x_t, t), \Sigma_\theta(x_t, t)). \tag{3}$$

In this process, the denoiser $\epsilon_\theta(x_t, t)$ learns to predict the added noise from the noisy image $x_t$ at a given time step $t$ (Song et al., 2020). During inference, random noise is sampled, and the denoiser iteratively removes it to generate a new image. By conditioning on a text prompt $c$, the denoiser $\epsilon_\theta(x_t, t, c)$ guides the generation process to align the resulting image with the provided text. Diffusion models demonstrate superior performance in understanding image details by generating high-quality images compared to other generative models (Dhariwal & Nichol, 2021). They can also be utilized for inpainting a specific region of an image. At each step, the untouched region is preserved by replacing it with the corresponding part of the original image, ensuring that only the specified region is generated and seamlessly blended with the rest of the image (Rombach et al., 2021).

## 4 Method

The final model, CLIPFUSION, is a fusion of our CLIP-based and diffusion-based models. For convenience, we call the CLIP-based model PatchCLIP (patch-based CLIP) and the diffusion-based model MapDiff (map-based diffusion). We first outline our feature extraction strategies from both models, tailored for anomaly detection.

**PatchCLIP**   CLIP processes an image by dividing it into multiple patches and outputs both patch embeddings and a class token embedding. While the class token embedding is aligned with text embeddings, patch embeddings can provide additional information, which we incorporate to enrich the spatial representation. Given that fine-grained details are handled by the diffusion-based model, PatchCLIP provides a simple yet effective spatial prior, thereby avoiding the cumbersome multi-scale window sliding in WinCLIP.

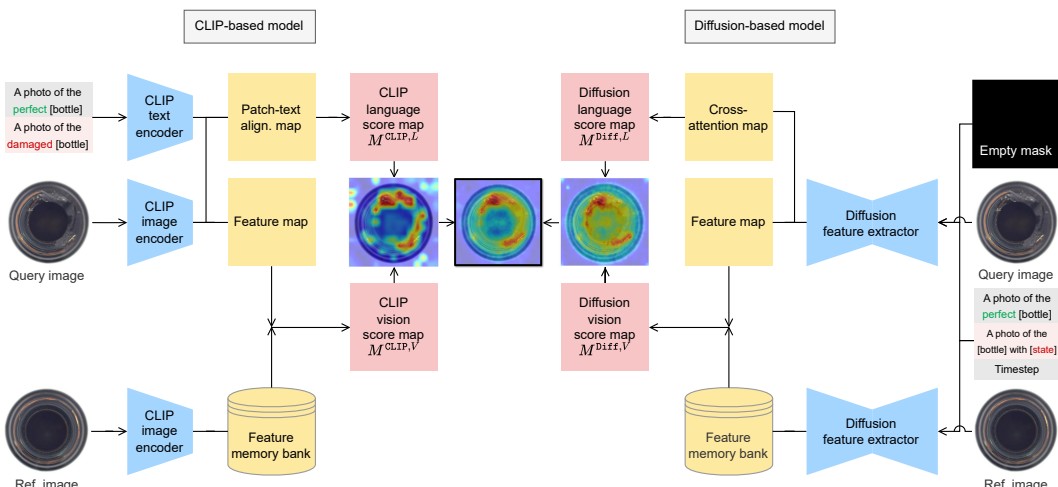

Figure 2: Overall framework for CLIPFUSION. The CLIP-based model and the diffusion-based model process query and reference images to generate anomaly maps. The final anomaly map is obtained by fusing the outputs of the vision and language components of both models.

Additionally, we utilize features extracted from CLIP's image encoder. To achieve this, we make use of the VV-attention mechanism proposed in CLIP surgery (Li et al., 2023b; 2024a). This mechanism was introduced to complement CLIP's QK-attention mechanism, which is specialized in global feature extraction. It facilitates local feature extraction by enabling direct interactions between values (hence the name VV-attention) that carry information about the patches themselves.

**MapDiff** We extract and utilize cross-attention maps and feature maps from the diffusion denoiser. To achieve this, we use an inpainting pipeline that takes a clean image as input, rather than a generation pipeline that starts with random noise. In the inpainting pipeline, a mask is provided to specify the region where noise will be added. Here, we use the trick of applying an empty mask. This prevents noise from being added, but the denoiser still perceives the original image as noisy, thereby allowing us to control the timestep and extract features at the desired level of detail.

By passing the clean image through the denoiser once, we extract the maps from its intermediate layers. A large timestep makes the denoiser interpret the image as highly noisy, capturing high-level pattern information, while a small timestep emphasizes finer details.

### 4.1 Anomaly Segmentation

The final anomaly score map $M \in \mathbb{R}^{H \times W}$ for segmentation is obtained by fusing the score maps from Patch-CLIP and MapDiff. In the zero-shot setup, only the language-guided score maps $M^{\texttt{CLIP},L}$ and $M^{\texttt{Diff},L}$ are used. In the few-shot setup, the vision-guided score maps $M^{\texttt{CLIP},V}$ and $M^{\texttt{Diff},V}$ are additionally incorporated. The final anomaly score map is computed as follows:

$$M := \alpha(M^{\texttt{CLIP},L} + M^{\texttt{CLIP},V}) + (1 - \alpha)(M^{\texttt{Diff},L} + M^{\texttt{Diff},V}), \tag{4}$$

where $\alpha$ is the weight between the models. In the zero-shot setup, $M := \alpha M^{\texttt{CLIP},L} + (1 - \alpha)M^{\texttt{Diff},L}$.

#### 4.1.1 Zero-shot

In the zero-shot setup, given a query image, we leverage the relationships between images and texts learned by multi-modal models to create anomaly score maps.

**CLIP Language-guided Score Map** $M^{\texttt{CLIP},L}$    The query image is input to the CLIP image encoder (e.g., ViT), which divides it into patches, processes them, and outputs patch embeddings. On the other hand, normal and abnormal text prompts are passed to the CLIP text encoder, producing normal and abnormal text embeddings. Then, the degree (Equation 1) to which each patch embedding is aligned to the abnormal text embedding is calculated for all patches and organized into a map. This is upsampled to the original image size through interpolation to obtain $M^{\texttt{CLIP},L}$.

**Diff Language-guided Score Map** $M^{\texttt{Diff},L}$    The query image is passed to the diffusion inpainting pipeline along with an empty mask, a timestep, and a text prompt. The text prompt includes an object word and a state word, for example:



"a photo of a [object] with [state]."



where [object] refers to the entity (e.g., "bottle") in the image, and [state] describes a possible abnormal condition (e.g., "crack") of it. The [state] token interacts with the image as it passes through the denoiser (e.g., U-Net), generating a cross-attention map. The cross-attention map of a diffusion model is a heatmap showing where and how strongly a text token is attended to across image locations. Because we use a token that denotes an anomalous state, the attention weight at a location serves as a proxy for anomaly likelihood. That is, a higher value at a specific location in the map indicates that the location responds more strongly to the [state] token (i.e., the abnormal condition). Therefore, we can use this cross-attention map as an anomaly score map (Figure 3). Additionally, we calculate the average of cross-attention maps derived using various possible abnormal conditions (e.g., "crack", "hole", "residue", "damage") to obtain $M^{\texttt{Diff},L}$.

This can be interpreted as follows. In anomaly segmentation, given an image $I = x$, we are interested in the probability that the $(h, w)$-th pixel is abnormal ($A_{h,w} = 1$). This probability can be expressed as follows by conditioning on a text prompt $T = c$:

$$p(A_{h,w} = 1 | I = x) = \sum_{c \in \mathcal{C}} p(A_{h,w} = 1 | I = x, T = c) p(T = c), \tag{5}$$

where $\mathcal{C}$ represents the set of text prompts that describe abnormal conditions. Assuming that the text prompt $T$ is uniform and interpreting $p(A_{h,w} = 1 | I = x, T = c)$ as the cross-attention map intensity at the corresponding location:

$$M^{\texttt{Diff},L}_{h,w} := \frac{1}{|\mathcal{C}|} \sum_{c \in \mathcal{C}} p(A_{h,w} = 1 | I = x, T = c). \tag{6}$$

### 4.1.2 Few-shot

In the few-shot setup, normal reference images are utilized. For the $k$-shot case, $k$ images are used to construct feature memory banks, which are then compared with the features of the query image. In general, we extract normal features from normal reference images by passing them through the CLIP image encoder and the diffusion denoiser. Then, the query image is passed through and the extracted features are compared with the normal features. The degree of deviation is used as an anomaly score.

**CLIP Vision-guided Score Map** $M^{\texttt{CLIP},V}$    The reference images are passed through the CLIP's image encoder to extract reference features and stored in the memory bank. Features with low-level semantic information are extracted from a middle encoder block, while those with high-level semantic information come from a later encoder block.

The set of reference features extracted from the encoder block $b$ is denoted as $\mathcal{R}^b$. For a given encoder block $b$, the feature map extracted from the query image is denoted as $F^b \in \mathbb{R}^{H \times W \times D_b}$. Then, $M^{\texttt{CLIP},V}_{h,w}$ is calculated by comparing the feature $F^b_{h,w}$ with the reference features $r \in \mathcal{R}^b$, as follows:

$$M^{\texttt{CLIP},V}_{h,w} := \frac{1}{|\mathcal{B}|} \sum_{b \in \mathcal{B}} \min_{r \in \mathcal{R}^b} \frac{1}{2} \left( 1 - \mathrm{sim}(F^b_{h,w}, r) \right), \tag{7}$$

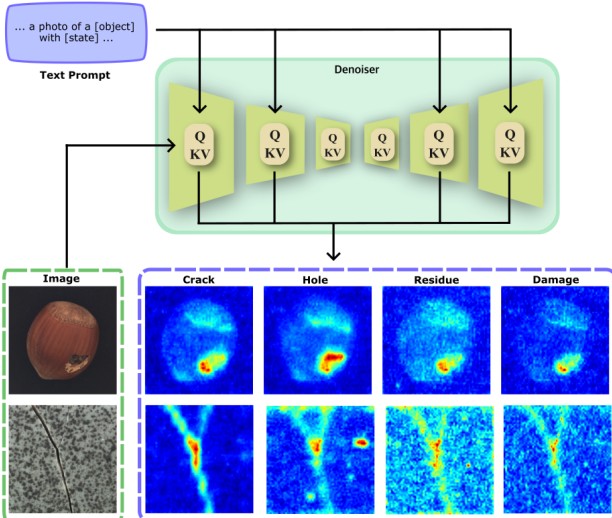

Figure 3: Examples of cross-attention maps extracted from a diffusion denoiser.

where $\mathcal{B}$ is the set of blocks $b$. That is, a score is computed based on the similarity, and the smallest among all reference features in $\mathcal{R}^b$ is conservatively used as the anomaly score for $b$. The final anomaly score is the average over $\mathcal{B}$.

**Diff Vision-guided Score Map $M^{\texttt{Diff},V}$** Reference images are passed through the diffusion denoiser to extract features, which are stored in the memory bank. The denoiser consists of three parts: an encoder (down-blocks), a bottleneck (mid-block), and a decoder (up-blocks). Information in the encoder is passed to the decoder via skip connections. The bottleneck is the most compressed stage, where much of the detailed information is lost, making it unsuitable for feature extraction. Therefore, we extract features from the decoder. Specifically, we focus on the middle and later decoder blocks, as they provide the most semantically rich features (Baranchuk et al., 2021; Zhao et al., 2023).

When timestep $t$ is given, the set of reference features extracted from the decoder block $b$ is denoted as $\mathcal{R}^{t,b}$. The feature map extracted from the query image at timestep $t$ and decoder block $b$ is denoted as $F^{t,b} \in \mathbb{R}^{H \times W \times D_b}$. Then, $M_{h,w}^{\texttt{Diff},V}$ is calculated as follows:

$$M_{h,w}^{\texttt{Diff},V} := \frac{1}{|\mathcal{T}|} \sum_{(t,b) \in \mathcal{T}} \min_{r \in \mathcal{R}^{t,b}} \frac{1}{2} \left( 1 - \text{sim}(F_{h,w}^{t,b}, r) \right), \tag{8}$$

where $\mathcal{T}$ is the set of combinations $(t, b)$.

### 4.2 Anomaly Classification

The final anomaly score $S \in \mathbb{R}$ for classification is obtained by fusing four scores:

$$S := \alpha(S^{\texttt{CLIP},L} + S^{\texttt{CLIP},V}) + (1 - \alpha)(S^{\texttt{Diff},L} + S^{\texttt{Diff},V}). \tag{9}$$

In the zero-shot setup, only the language-guided scores are used. Specifically, the CLIP language-guided score $S^{\texttt{CLIP},L}$ is defined as the degree (Equation 1) of alignment between the query image's class token embedding and the abnormal text embedding. The diffusion language-guided score $S^{\texttt{Diff},L}$ aggregates the diffusion language-guided score map $M^{\texttt{Diff},L}$ and is defined as follows:

$$S^{\texttt{Diff},L} := 1 - \text{median}(M^{\texttt{Diff},L}) / \max(M^{\texttt{Diff},L}). \tag{10}$$

This means that the score becomes larger when the degree of focus of the cross-attention map is high, that is, when the maximum is significantly greater than the median.

In the few-shot setup, we additionally aggregate the vision-guided score maps, defining the scores as follows in accordance with common practice (Jeong et al., 2023):

$$S^{\texttt{CLIP},V} := \max(M^{\texttt{CLIP},V}), \ S^{\texttt{Diff},V} := \max(M^{\texttt{Diff},V}). \tag{11}$$

## 5 Comparisons

**Datasets**  We use the MVTec-AD (Bergmann et al., 2019) and VisA (Zou et al., 2022) datasets, which are mainly used as benchmark datasets in anomaly detection. Each dataset consists of several object categories, and each object category is divided into a training set containing only normal images and a test set containing a mixture of normal and abnormal images. Query images are from the test set, and reference images are sampled from the training set.

**Baseline methods**  We compare against methods with the same setup, that is, methods that use only the provided zero-shot or few-shot normal images and do not rely on any additional supervision. This setup reflects a realistic and challenging scenario commonly encountered in practice.

Specifically, we compare our method, CLIPFUSION, with methods utilizing vision models (PaDiM, Patch-Core) and those utilizing vision-language models (CLIP-AC, Trans-MM, MaskCLIP, WinCLIP, PromptAD[3]). Note that, among these methods, only CLIPFUSION and WinCLIP are fully capable of handling both segmentation and classification tasks in both zero-shot and few-shot scenarios.

**Experimental details**  For PatchCLIP, we employ OpenCLIP (Ilharco et al., 2021; Cherti et al., 2023; Schuhmann et al., 2022), which has been pretrained on LAION-400M (Schuhmann et al., 2021), using a Vision Transformer (ViT) (Dosovitskiy, 2020) as its image encoder. For MapDiff, we utilize the pretrained Stable Diffusion v2 model (Rombach et al., 2022), where the denoiser is based on U-Net (Ronneberger et al., 2015).

We set $\alpha = 0.25$ in Equation 4 for segmentation and $\alpha = 0.75$ in Equation 9 for classification to reflect the different importance of the models in each task. We report the average performance and standard deviations over five runs with different seeds. Further details are in Appendix A.2.

### 5.1 Anomaly Segmentation

Table 1 presents a comparison of our method's performance with baseline methods for anomaly segmentation. The results show that CLIPFUSION consistently outperforms the baseline methods across all setups, datasets, and metrics. Our method, in particular, demonstrates dramatic performance improvements in the zero-shot setup. This highlights the critical contribution of the diffusion-based model's cross-attention map to zero-shot anomaly segmentation. Additionally, the significant performance improvements in the few-shot setup demonstrate how the diffusion-based model complements the CLIP-based model.

### 5.2 Anomaly Classification

Table 2 shows the performance comparison between our method and baseline methods for anomaly classification. CLIPFUSION consistently outperforms all baseline methods. This indicates that the diffusion-based model enhances performance not only in the segmentation task but also in the classification task by reinforcing local information.

In both segmentation and classification tasks, the consistent performance improvements as the number of shots increase highlight the scalability of our method with minimal additional data. Furthermore, the smaller standard deviations observed in our results indicate the robustness of our approach. This robustness can be attributed to the integration of multiple models, which mitigates the variability.

---

[3]The authors of PromptAD use the test set as a validation set during training to record the best performance for each category separately and report the aggregated results. For a fair comparison, we treat the training epoch as a fixed hyperparameter across categories and report the best-performing checkpoint reproduced using the official codebase. See Appendix A.5 for more details.

Table 1: Quantitative results for anomaly segmentation.

| Setup | Method | MVTec-AD | | VisA | |
|---|---|---|---|---|---|
| | | AUROC | AUPRO | AUROC | AUPRO |
| 0-shot | Trans-MM (Chefer et al., 2021) | 57.5±0.0 | 21.9±0.0 | 49.4±0.0 | 10.2±0.0 |
| | MaskCLIP (Zhou et al., 2022) | 63.7±0.0 | 40.5±0.0 | 60.9±0.0 | 27.3±0.0 |
| | WinCLIP (Jeong et al., 2023) | 85.1±0.0 | 64.6±0.0 | 79.6±0.0 | 56.8±0.0 |
| | **CLIPFUSION (ours)** | **90.9±0.0** | **87.1±0.0** | **92.1±0.0** | **83.3±0.0** |
| 1-shot | PaDiM (Defard et al., 2021) | 89.3±0.9 | 73.3±2.0 | 89.9±0.8 | 64.3±2.4 |
| | PatchCore (Roth et al., 2022) | 92.0±1.0 | 79.7±2.0 | 95.4±0.6 | 80.5±2.5 |
| | WinCLIP (Jeong et al., 2023) | 95.2±0.5 | 87.1±1.2 | 96.4±0.4 | 85.1±2.1 |
| | PromptAD (Li et al., 2024a) | 94.9±0.8 | 87.7±1.5 | 95.7±0.9 | 84.0±1.2 |
| | **CLIPFUSION (ours)** | **95.8±0.1** | **90.4±0.1** | **97.3±0.1** | **87.0±0.3** |
| 2-shot | PaDiM (Defard et al., 2021) | 91.3±0.7 | 78.2±1.8 | 92.0±0.7 | 70.1±2.6 |
| | PatchCore (Roth et al., 2022) | 93.3±0.6 | 82.3±1.3 | 96.1±0.5 | 82.6±2.3 |
| | WinCLIP (Jeong et al., 2023) | 96.0±0.3 | 88.4±0.9 | 96.8±0.3 | 86.2±1.4 |
| | PromptAD (Li et al., 2024a) | 95.0±0.7 | 88.3±1.0 | 96.3±0.9 | 84.8±0.8 |
| | **CLIPFUSION (ours)** | **96.3±0.1** | **91.4±0.2** | **97.7±0.1** | **87.3±0.2** |
| 4-shot | PaDiM (Defard et al., 2021) | 92.6±0.7 | 81.3±1.9 | 93.2±0.5 | 72.6±1.9 |
| | PatchCore (Roth et al., 2022) | 94.3±0.5 | 84.3±1.6 | 96.8±0.3 | 84.9±1.4 |
| | WinCLIP (Jeong et al., 2023) | 96.2±0.3 | 89.0±0.8 | 97.2±0.2 | 87.6±0.9 |
| | PromptAD (Li et al., 2024a) | 95.5±0.6 | 89.0±0.6 | 96.3±0.8 | 85.1±0.9 |
| | **CLIPFUSION (ours)** | **96.8±0.1** | **92.1±0.2** | **98.0±0.1** | **87.7±0.5** |

Table 2: Quantitative results for anomaly classification.

| Setup | Method | MVTec-AD | | VisA | |
|---|---|---|---|---|---|
| | | AUROC | AUPR | AUROC | AUPR |
| 0-shot | CLIP-AC (Radford et al., 2021) | 74.0±0.0 | 89.1±0.0 | 59.3±0.0 | 67.0±0.0 |
| | WinCLIP (Jeong et al., 2023) | 91.8±0.0 | 96.5±0.0 | 78.1±0.0 | 81.2±0.0 |
| | **CLIPFUSION (ours)** | **93.6±0.0** | **97.0±0.0** | **79.5±0.0** | **83.6±0.0** |
| 1-shot | PaDiM (Defard et al., 2021) | 76.6±3.1 | 88.1±1.7 | 62.8±5.4 | 68.3±4.0 |
| | PatchCore (Roth et al., 2022) | 83.4±3.0 | 92.2±1.5 | 79.9±2.9 | 82.8±2.3 |
| | WinCLIP (Jeong et al., 2023) | 93.1±2.0 | 96.5±0.9 | 83.8±4.0 | 85.1±4.0 |
| | PromptAD (Li et al., 2024a) | 92.6±1.0 | 95.7±0.9 | 83.1±2.1 | 85.7±1.2 |
| | **CLIPFUSION (ours)** | **95.4±0.8** | **97.8±0.4** | **86.5±0.1** | **88.4±0.2** |
| 2-shot | PaDiM (Defard et al., 2021) | 78.9±3.1 | 89.3±1.7 | 67.4±5.1 | 71.6±3.8 |
| | PatchCore (Roth et al., 2022) | 86.3±3.3 | 93.8±1.7 | 81.6±4.0 | 84.8±3.2 |
| | WinCLIP (Jeong et al., 2023) | 94.4±1.3 | 97.0±0.7 | 84.6±2.4 | 85.8±2.7 |
| | PromptAD (Li et al., 2024a) | 93.2±1.0 | 96.1±1.0 | 85.0±1.8 | 87.2±1.8 |
| | **CLIPFUSION (ours)** | **96.3±0.3** | **98.2±0.2** | **87.1±0.5** | **88.8±0.5** |
| 4-shot | PaDiM (Defard et al., 2021) | 80.4±2.5 | 90.5±1.6 | 72.8±2.9 | 75.6±2.2 |
| | PatchCore (Roth et al., 2022) | 88.8±2.6 | 94.5±1.5 | 85.3±2.1 | 87.5±2.1 |
| | WinCLIP (Jeong et al., 2023) | 95.2±1.3 | 97.3±0.6 | 87.3±1.8 | 88.8±1.8 |
| | PromptAD (Li et al., 2024a) | 93.8±0.7 | 96.5±0.7 | 86.2±1.6 | 88.1±1.9 |
| | **CLIPFUSION (ours)** | **96.9±0.2** | **98.6±0.1** | **88.1±0.2** | **89.7±0.1** |

# 6 Empirical Study

In this section, results are reported under the one-shot setup.

Table 3: Comparison of different models.

| Task | Model | | Dataset | |
|------|-------|------|---------|------|
| | CLIP | Diff | MVTec-AD | VisA |
| Seg. | ✓ | × | 94.7 | 93.7 |
| | × | ✓ | 95.1 | 96.5 |
| | ✓ | ✓ | **95.8** | **97.3** |
| Cls. | ✓ | × | 93.5 | 82.2 |
| | × | ✓ | 87.2 | 78.9 |
| | ✓ | ✓ | **95.4** | **86.5** |

Table 4: Comparison of different modalities.

| Task | Modality | | Dataset | |
|------|----------|--------|---------|------|
| | Language | Vision | MVTec-AD | VisA |
| Seg. | ✓ | × | 90.9 | 92.1 |
| | × | ✓ | 95.4 | 96.1 |
| | ✓ | ✓ | **95.8** | **97.3** |
| Cls. | ✓ | × | 93.2 | 79.5 |
| | × | ✓ | 90.5 | 82.8 |
| | ✓ | ✓ | **95.4** | **86.5** |

Table 5: Timestep and block configurations.

| Task | Configuration | MVTec-AD | VisA |
|------|---------------|----------|------|
| Seg. | Step ↑, Block ↑ | 94.3 | 95.1 |
| | Step ↑, Block ↓ | **95.1** | **96.5** |
| Cls. | Step ↑, Block ↑ | 86.3 | 76.2 |
| | Step ↑, Block ↓ | **87.2** | **78.9** |

Table 6: Decoder blocks for feature maps.

| Task | Early Block | MVTec-AD | VisA |
|------|-------------|----------|------|
| Seg. | w/ | 94.9 | 93.7 |
| | w/o | **95.1** | **96.5** |
| Cls. | w/ | 86.5 | 74.9 |
| | w/o | **87.2** | **78.9** |

## 6.1 Ablation Study

Table 3 presents the comparative performance of different models. As expected, the diffusion-based model, which captures local features effectively, performs better in the segmentation task, while the CLIP-based model, which captures global features effectively, performs better in the classification task. The combination of different models consistently yields better results compared to using either model alone. This empirical evidence suggests that the two models capture complementary features, making their integration beneficial for both segmentation and classification tasks.

Table 4 presents the performance of different modalities, highlighting their complementary strengths through the combination of semantic and spatial information.

## 6.2 Extraction Strategies from Diffusion Denoisers

We explore strategies to effectively extract features from diffusion denoisers. Here, we conduct experiments using only MapDiff to see the effect more clearly.

Table 5 highlights the impact of matching timestep and block numbers when calculating $M^{\texttt{Diff},V}$ (Equation 8). Note that the decoder consists of four blocks (Block 0 to Block 3). Matching higher timesteps with lower blocks yields better performance, and vice versa (Timestep 201 → Block 3, Timestep 401 → Block 2, Timestep 801 → Block 1). This pattern reflects the intrinsic behavior of the diffusion denoiser: at higher timesteps, the denoiser interprets the input image as highly noisy and prioritizes the extraction of global features. The lower blocks, which process coarse-grained information, are particularly well-suited for this task. Using higher timesteps can be seen as a regularization strategy, as they suppress local details and highlight broader patterns.

Table 6 shows that when calculating $M^{\texttt{Diff},V}$ (Equation 8), including the early block (Block 0) does not enhance the model's performance. On the contrary, excluding this block consistently yields better results. This suggests that the early stages of the decoder act as preparatory steps for the later stages, without forming concrete features.

## 6.3 Inference speed

Our method can be viewed as an extension of WinCLIP (Jeong et al., 2023). We report the inference latency measured on an NVIDIA A100 GPU (40 GB) in Table 7, with a batch size of 1. As shown, our method is approximately 6–8 times faster than WinCLIP.

Table 7: Comparison of latency (ms).

| Latency [ms] | 0-shot | 1-shot | 2-shot | 4-shot |
|---|---|---|---|---|
| WinCLIP | 4782 | 9514 | 9633 | 9710 |
| CLIPFUSION | **783** | **1240** | **1269** | **1278** |

To capture fine-grained details, WinCLIP slides multi-scale windows over the query image, resulting in hundreds of forward passes. In contrast, our CLIP-based model utilizes patch embeddings obtained from a single forward pass. On top of this, we employ a diffusion-based model not as a denoiser for iterative generation, but as a feature extractor to capture fine-grained details. As a result, our method requires only a few forward passes in total.

### 6.4 One-for-all paradigm

IIPAD (Lv et al.) recently proposes a new setting for few-shot anomaly detection, termed the one-for-all paradigm. It aims to train a single unified model that can detect anomalies across multiple object categories. In this paradigm, the one-shot setting refers to using $N$ normal images (one for each category), where $N$ is the number of categories. Apart from this, we follow the same experimental setting as described in Section 5. Table 8 presents a comparison with IIPAD (the SOTA method under this paradigm) on the MVTec-AD dataset in the one-shot setting, where our method achieves higher performance. See Appendix A.4 for more results.

Table 8: One-for-all paradigm results.

| Method | AUROC | AUPR |
|---|---|---|
| IIPAD | 94.2 | 97.2 |
| CLIPFUSION | **94.5** | **97.3** |

## 7 Conclusion

We propose a methodology that synergistically combines discriminative and generative foundation models, leveraging their complementary strengths. As a concrete application, we introduce CLIPFUSION, which integrates CLIP and diffusion models for anomaly detection and consistently outperforms baselines in segmentation and classification across zero-shot and few-shot scenarios. This holistic use of features from discriminative and generative models may benefit the research community in tackling complex recognition tasks.

### Acknowledgments

This work was conducted by Center for Applied Research in Artificial Intelligence(CARAI) grant funded by Defense Acquisition Program Administration(DAPA) and Agency for Defense Development(ADD) (UD230017TD).

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

# A   Appendix

## A.1   Further motivation for feature-based anomaly detection

Addressing anomaly detection with a single-perspective approach is challenging due to the open-ended nature of abnormality. Discriminative models focus on global features but often miss fine-grained details, which are crucial for detecting local structural or textural anomalies.

When utilizing generative models in anomaly detection, the reconstruction-based approach has traditionally been the dominant paradigm (Zavrtanik et al., 2021; Zhang et al., 2023a; Guo et al., 2023). This approach compares the original image and the reconstructed image at both ends. However, since generative models are primarily designed for creative image generation, they can produce high reconstruction errors even for normal images. In addition, deciding how to apply the mask, including its location and size is challenging.[4] As a result, the feature-based approach has recently gained prominence.

Building on this shift, we propose that a more fundamental approach to utilizing generative models for anomaly detection is to extract features midway through the process. This leverages the image understanding capability that underlies their image generation capability.

### A.2 Further experimental details

**Data pre-processing** For PatchCLIP, we use the same pre-processing pipeline as in WinCLIP (Jeong et al., 2023), which is specified in OpenCLIP (Ilharco et al., 2021), for both MVTec-AD and VisA datasets. First, bicubic interpolation is used to resize images to a resolution of 240 to fit the ViT-B-16-plus-240 model. After which, channel-wise standardization is applied with the precomputed values of (0.48145466, 0.4578275, 0.40821073) for the mean and (0.26862954, 0.26130258, 0.27577711) for the standard deviation. For MapDiff, we use bilinear interpolation to resize images to a resolution of 512 before feeding into a customized Stable Diffusion Inpainting pipeline from the diffusers library (von Platen et al., 2022).

**Prompts** The prompts are derived from those used to train CLIP for the ImageNet dataset (Deng et al., 2009). For example, in the case of a segmentation task on MVTec-AD, PatchCLIP uses the prompt "`a good, cropped picture of the [state] [object] for classification`", where [state] is either "`damaged`" or "`perfect`". MapDiff uses the prompt "`a close-up cropped png photo of a [object] with [state] for anomaly segmentation`" for query images. The [state] includes commonly observed anomaly types "`crack`", "`hole`", and "`residue`", along with a generic descriptor "`damage`". For reference images, it uses the prompt "`a photo of a perfect [object]`".

**Evaluation metrics** We follow previous studies (Cohen & Hoshen, 2020; Defard et al., 2021; Roth et al., 2022; Jeong et al., 2023) and use the following performance metrics. For anomaly classification, we report the Area Under the ROC Curve (AUROC) at the image level and the Area Under the Precision-Recall Curve (AUPR). For anomaly segmentation, we report the AUROC at the pixel level and the Area Under the Per-Region Overlap Curve (AUPRO) (Bergmann et al., 2020; Li et al., 2021). Both AUPR and AUPRO are particularly sensitive to class imbalance.

**Hyperparameters** We compare weights of (1, 0), (0, 1), (0.75, 0.25), (0.25, 0.75), and (0.5, 0.5) between the CLIP and diffusion models. We observe a consistent trend: (0.25, 0.75) is effective for segmentation tasks, while (0.75, 0.25) performs better for classification tasks. The same weights are used across all datasets for each task. Please refer to Table 3.

**Baseline performance** The performance of Trans-MM, MaskCLIP, PaDiM, PatchCore, CLIP-AC, and WinCLIP are referenced from the WinCLIP paper (Jeong et al., 2023).

### A.3 Further empirical study

Table 9 demonstrates the effectiveness of extracting cross-attention maps from the encoder (down-blocks) and decoder (up-blocks) while excluding the bottleneck (mid-block). Integrating the bottleneck results in suboptimal performance due to its lack of spatial granularity critical for tasks like anomaly detection.

Table 10 illustrates the impact of using state ensemble for a cross-attention map. Without ensemble, only a generic term "`damage`" is used to generate the cross-attention map. In contrast, the ensemble approach

---

[4]Since the location of the anomalous part is not known, it is unclear where to apply the mask. Also, if the mask is too large, the reconstructed image may deviate significantly from the original image. Conversely, if the mask is too small and fails to fully cover the anomaly, the anomaly may persist or even extend in the reconstructed image.

Table 9: Denoiser components for cross-attention maps.

| Task | Denoiser Component | | | Dataset | |
|------|-----|-------|-----|----------|------|
|      | Enc | Bneck | Dec | MVTec-AD | VisA |
| Seg. | × | × | ✓ | 94.4 | 96.0 |
|      | ✓ | ✓ | ✓ | 94.4 | 96.4 |
|      | ✓ | × | ✓ | **95.1** | **96.5** |
| Cls. | × | × | ✓ | 83.2 | 74.1 |
|      | ✓ | ✓ | ✓ | 86.8 | 76.1 |
|      | ✓ | × | ✓ | **87.2** | **78.9** |

averages cross-attention maps derived from both the generic term "`damage`" and more specific terms such as "`crack`," "`hole`," and "`residue`." This approach significantly improves performance across tasks. By incorporating these multiple specific states, the method generates diverse cross-attention maps that capture fine-grained and complementary aspects of anomalies.

Table 10: State ensemble (generic and specific).

| Task | Ensemble | Dataset | |
|------|----------|----------|------|
|      |          | MVTec-AD | VisA |
| Seg. | w/o | 94.0 | 95.8 |
|      | w/  | **95.1** | **96.5** |
| Cls. | w/o | 85.4 | 73.1 |
|      | w/  | **87.2** | **78.9** |

Table 11 shows the impact of customizing state descriptors for each object class on performance. For example, for both Cable and Pill, replacing the default set of states with a class-specific set composed of frequently occurring anomaly types leads to consistent performance improvements.

Table 11: Effect of state customization.

| Object | State Set | AUROC |
|--------|-----------|-------|
| Cable | "`crack`", "`hole`", "`residue`", "`damage`" | 94.6 |
|       | "`crack`", "`poke`", "`scratch`" | **95.0** |
| Pill  | "`crack`", "`hole`", "`residue`", "`damage`" | 83.0 |
|       | "`crack`", "`scratch`", "`residue`" | **83.5** |

Table 12 presents results obtained with different sets of states. The results show our method remains reasonably robust across these variations.

Table 12: Different sets of states.

| States | Task | Dataset | |
|--------|------|----------|------|
|        |      | MVTec-AD | VisA |
| "`contamination`", "`bend`", "`cut`", "`damage`" | Seg. | 95.8 | 97.0 |
|        | Cls. | 95.4 | 85.9 |
| "`break`", "`fold`", "`stain`", "`damage`" | Seg. | 95.8 | 97.3 |
|        | Cls. | 95.6 | 86.1 |

## A.4 Further results in the one-for-all paradigm

We evaluate anomaly detection performance under the one-for-all paradigm, where a single model is applied across all object categories. Recall that, in this paradigm, the one-shot setting provides one image per

category, for a total of $N$ images across $N$ object categories. IIPAD (Lv et al.) trains a prompt generator using these examples for the one-for-all setting. In contrast, CLIPFUSION requires no additional training. Nevertheless, it outperforms or matches IIPAD across all configurations and datasets for both segmentation and classification, as shown in Tables 13 and 14.

Table 13: Quantitative results for anomaly segmentation in the one-for-all paradigm.

| Setup | Method | MVTec-AD | | VisA | |
|-------|--------|----------|--------|-------|--------|
| | | AUROC | AUPR | AUROC | AUPR |
| 1-shot | IIPAD | **96.4** | 89.8 | 96.9 | **87.3** |
| | CLIPFUSION | 96.0 | **91.1** | **97.2** | **87.3** |
| 2-shot | IIPAD | **96.7** | 90.3 | 97.2 | 87.9 |
| | CLIPFUSION | 96.4 | **91.9** | **97.4** | **88.1** |
| 4-shot | IIPAD | **97.0** | 91.2 | 97.4 | 88.3 |
| | CLIPFUSION | 96.8 | **92.5** | **98.1** | **88.6** |

Table 14: Quantitative results for anomaly classification in the one-for-all paradigm.

| Setup | Method | MVTec-AD | | VisA | |
|-------|--------|----------|--------|-------|--------|
| | | AUROC | AUPR | AUROC | AUPR |
| 1-shot | IIPAD | 94.2 | 97.2 | 85.4 | 87.5 |
| | CLIPFUSION | **94.5** | **97.3** | **85.5** | **88.1** |
| 2-shot | IIPAD | 95.7 | **97.9** | 86.7 | 88.6 |
| | CLIPFUSION | **96.0** | **97.9** | **86.8** | **89.3** |
| 4-shot | IIPAD | 96.1 | 98.1 | **88.3** | 89.6 |
| | CLIPFUSION | **96.4** | **98.2** | **88.3** | **90.0** |

## A.5 Limitations of few-shot training

Training-based methods inherently suffer from epoch selection sensitivity. Figure 4a shows that the epoch at which PromptAD performs best differs significantly across object categories. In conventional training setups, such epoch selection is guided by a validation set. However, in few-shot anomaly detection, where only one to four images are available, it is practically hard to reserve any for a separate validation set. The authors of PromptAD circumvent this by using the test set for validation in their official implementation.[5] However, test sets are not accessible during model training in real-world applications. In contrast, CLIPFUSION requires no additional training and uses few-shot examples solely for feature extraction, making validation sets and epoch selection unnecessary.

Figure 4b reveals that even when the training epoch is fixed, performance varies considerably across different random seeds. This highlights the instability and seed sensitivity of few-shot training, making it difficult to ensure consistent and reproducible results without a validation set.

## A.6 Effect of weights between the CLIP and diffusion models

We analyze the effect of varying the weights assigned to the CLIP and diffusion models. Recall that $\alpha$ and $(1-\alpha)$ denote the weights of the CLIP and the diffusion model, respectively. By changing $\alpha$, we adjust their relative contributions and evaluate the performance. As shown in Table 15, segmentation achieves the best performance among the tested alpha values at $\alpha = 0.25$, while classification performed best at $\alpha = 0.75$.

---

[5]https://github.com/FuNz-0/PromptAD

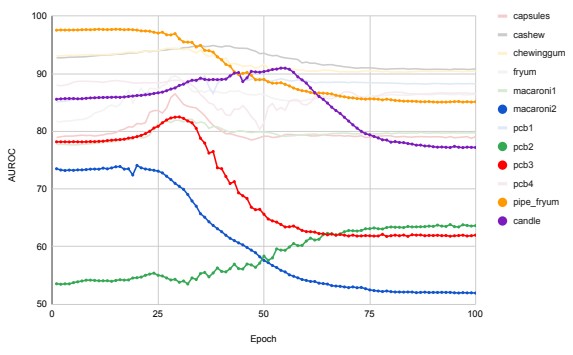

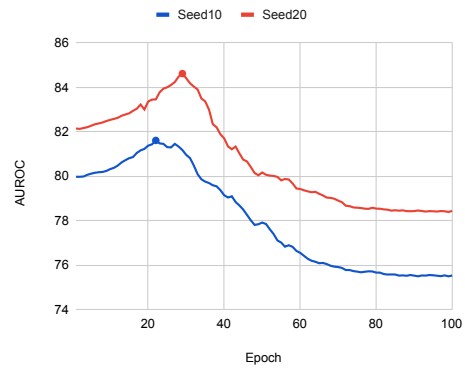

(a) AUROC for individual objects.

(b) Average AUROC over all objects.

Figure 4: AUROC across training epochs on the VisA dataset (1-shot setting).

Table 15: Effect of weights on performance.

| Task | Weight | | Dataset | |
|------|--------|------|----------|------|
| | **CLIP** | **Diff** | **MVTec-AD** | **VisA** |
| | 1 | 0 | 94.7 | 93.7 |
| | 0.75 | 0.25 | 95.4 | 96.1 |
| Seg. | 0.5 | 0.5 | 95.7 | 97.2 |
| | 0.25 | 0.75 | **95.8** | **97.3** |
| | 0 | 1 | 95.1 | 96.5 |
| | 1 | 0 | 93.5 | 82.2 |
| | 0.75 | 0.25 | **95.4** | **86.5** |
| Cls. | 0.5 | 0.5 | 95.0 | 86.4 |
| | 0.25 | 0.75 | 92.7 | 83.9 |
| | 0 | 1 | 87.2 | 78.9 |

### A.7 Failure cases

Figure 5 provides representative failure cases. In Figure 5(a), although the anomaly lies in the outer sheath of the cables, the diffusion-based model over-activates on the dense and repetitive wire patterns, failing to highlight the true defect. Figure 5(b) shows a logical anomaly, where the absence of a transistor itself constitutes the defect. Lacking this contextual understanding, the models struggle to determine where to attend, resulting in mislocalization.

### A.8 Results on different backbones

Table 16 reports the performance of our framework with different diffusion backbones. We observe that the results are consistent across backbones. This indicates that our method is robust to the choice of diffusion backbone.

### A.9 Pseudocode

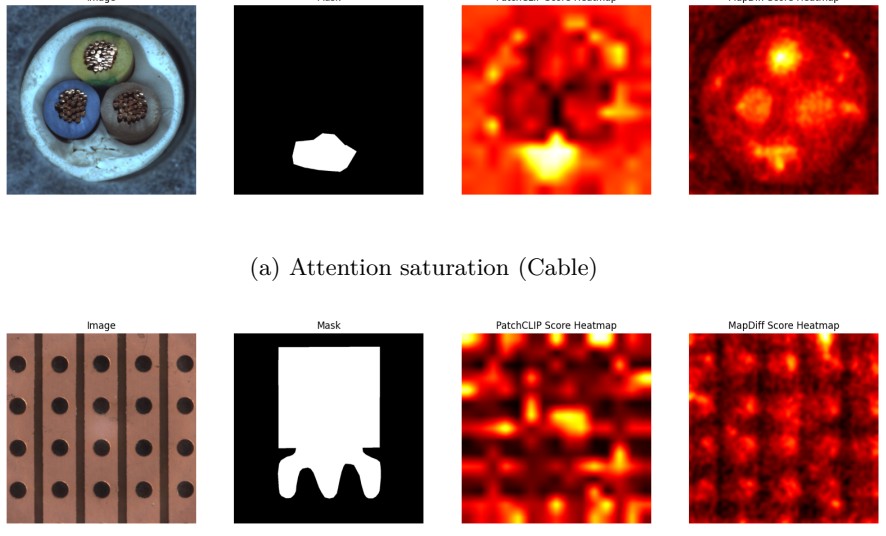

(a) Attention saturation (Cable)

(b) Logical anomaly (Transistor)

Figure 5: Failure analysis examples.

Table 16: Different backbones.

| Backbone | Task | Dataset | |
|---|---|---|---|
| | | MVTec-AD | VisA |
| Stable Diffusion v1.5 | Seg. | 95.8 | 96.7 |
| | Cls. | 93.5 | 86.4 |
| Stable Diffusion v2.0 | Seg. | 95.8 | 97.3 |
| | Cls. | 95.4 | 86.5 |
| Stable Diffusion v2.1 | Seg. | 95.8 | 97.3 |
| | Cls. | 94.6 | 86.5 |

---

**Algorithm 1** CLIPFUSION for Anomaly **Segmentation**

---

**Require:** Query image $x$; optional $k$ normal reference images $\mathcal{R} = \{r_i\}_{i=1}^k$; object label $o$; state set $\mathcal{C}$; CLIP image and text encoders $f_I, f_T$; diffusion denoiser $\epsilon_\theta$; CLIP blocks $\mathcal{B}$; Diffusion timestep-block pairs $\mathcal{T} = \{(t, b)\}$; fusion weight $\alpha \in [0, 1]$
**Ensure:** Pixel-level anomaly map $M$

1: Initialize $M^{\texttt{CLIP},L}, M^{\texttt{Diff},L} \leftarrow 0$              $\triangleright$ See Sec. 4.1.1 zero-shot
2: **(CLIP, language-guided)**
3: Encode $x$ with $f_I$ to obtain the patch embeddings
4: Encode prompts "a photo of the perfect $[o]$" and "a photo of the damaged $[o]$" with $f_T$
5: Form $M^{\texttt{CLIP},L}$ by per-patch alignment (abnormal vs. normal)
6: **(Diffusion, language-guided)**
7: **for** state $s \in \mathcal{C}$ **do**
8:      Run $\epsilon_\theta$ on $x$ with empty mask and prompt "a photo of a $[o]$ with $[s]$"
9:      Extract cross-attention map for state $s$; accumulate $M^{\texttt{Diff},L} \leftarrow M^{\texttt{Diff},L} + \text{CrossAttn}(s)$
10: **end for**
11: $M^{\texttt{Diff},L} \leftarrow M^{\texttt{Diff},L}/|\mathcal{C}|$

12: Initialize $M^{\texttt{CLIP},V}, M^{\texttt{Diff},V} \leftarrow 0$              $\triangleright$ See Sec. 4.1.2 few-shot
13: **if** $k > 0$ **then**
14:      **(CLIP, vision-guided)**
15:      For each $r \in \mathcal{R}$ and $b \in \mathcal{B}$, extract features from $f_I$ and store into $\mathcal{R}^b$
16:      For each $(h, w)$,
17:          $M_{h,w}^{\texttt{CLIP},V} \leftarrow \frac{1}{|\mathcal{B}|} \sum_{b\in\mathcal{B}} \min_{r\in\mathcal{R}^b} \frac{1}{2}\left(1 - \text{sim}(F_{h,w}^b, r)\right)$        (Eq. 7)
18:      **(Diffusion, vision-guided)**
19:      For each $r \in \mathcal{R}$ and $(t, b) \in \mathcal{T}$, extract features from $\epsilon_\theta$ and store into $\mathcal{R}^{t,b}$
20:      For each $(h, w)$,
21:          $M_{h,w}^{\texttt{Diff},V} \leftarrow \frac{1}{|\mathcal{T}|} \sum_{(t,b)\in\mathcal{T}} \min_{r\in\mathcal{R}^{t,b}} \frac{1}{2}\left(1 - \text{sim}(F_{h,w}^{t,b}, r)\right)$        (Eq. 8)
22: **end if**
23: **Fuse maps** $M \leftarrow \alpha(M^{\texttt{CLIP},L} + M^{\texttt{CLIP},V}) + (1-\alpha)(M^{\texttt{Diff},L} + M^{\texttt{Diff},V})$        (Eq. 4)
24: **return** $M$

---

---

**Algorithm 2** CLIPFUSION for Anomaly **Classification**

---

**Require:** Query image $x$; optional $k$ normal reference images $\mathcal{R} = \{r_i\}_{i=1}^k$; object label $o$; state set $\mathcal{C}$; CLIP image and text encoders $f_I, f_T$; diffusion denoiser $\epsilon_\theta$; CLIP blocks $\mathcal{B}$; Diffusion timestep–block pairs $\mathcal{T} = \{(t, b)\}$; fusion weight $\alpha \in [0, 1]$

**Ensure:** Image-level anomaly score $S$

1: Initialize $S^{\texttt{CLIP},L}, S^{\texttt{Diff},L} \leftarrow 0$                                    ▷ See Sec. 4.2
2: **(CLIP, language-guided)**
3: Encode $x$ with $f_I$ to obtain the class token embedding
4: Encode prompts "a photo of the perfect $[o]$" and "a photo of the damaged $[o]$" with $f_T$
5: Calculate $S^{\texttt{CLIP},L}$ via Eq. (1)
6: **(Diffusion, language-guided)**
7: Compute $M^{\texttt{Diff},L}$ as in Alg. 1
8: $S^{\texttt{Diff},L} \leftarrow 1 - \text{median}(M^{\texttt{Diff},L})/\max(M^{\texttt{Diff},L})$                                (Eq. 10)

9: Initialize $S^{\texttt{CLIP},V}, S^{\texttt{Diff},V} \leftarrow 0$
10: **if** $k > 0$ **then**
11:     **(CLIP, vision-guided)**
12:     Build memory banks $\mathcal{R}^b$ from $f_I$; compute $M^{\texttt{CLIP},V}$ as in Alg. 1
13:     $S^{\texttt{CLIP},V} \leftarrow \max(M^{\texttt{CLIP},V})$                                            (Eq. 11)
14:     **(Diffusion, vision-guided)**
15:     Build memory banks $\mathcal{R}^{t,b}$ from $\epsilon_\theta$; compute $M^{\texttt{Diff},V}$ as in Alg. 1
16:     $S^{\texttt{Diff},V} \leftarrow \max(M^{\texttt{Diff},V})$                                         (Eq. 11)
17: **end if**
18: **Fuse scores** $S \leftarrow \alpha\big(S^{\texttt{CLIP},L} + S^{\texttt{CLIP},V}\big) + (1-\alpha)\big(S^{\texttt{Diff},L} + S^{\texttt{Diff},V}\big)$          (Eq. 9)
19: **return** $S$

---

## A.10 Detailed quantitative results

Table 17: AUROC (%) results on MVTec-AD classification.

| Category | k=0 | | k=1 | | | | k=2 | | | | k=4 | | | |
|---|---|---|---|---|---|---|---|---|---|---|---|---|---|---|
| | WinCLIP | CLIPFUSION | PaDiM | PatchCore | WinCLIP | CLIPFUSION | PaDiM | PatchCore | WinCLIP | CLIPFUSION | PaDiM | PatchCore | WinCLIP | CLIPFUSION |
| Bottle | 99.2±0.0 | 100.0±0.0 | 97.4±0.7 | 99.4±0.4 | 98.2±0.0 | 100.0±0.0 | 98.5±1.0 | 99.2±0.3 | 99.3±0.3 | 100.0±0.0 | 98.8±0.2 | 99.2±0.3 | 99.3±0.4 | 100.0±0.0 |
| Cable | 86.5±0.0 | 87.3±0.0 | 57.7±4.6 | 88.8±4.2 | 88.9±1.9 | 95.0±0.9 | 62.3±5.9 | 91.0±2.7 | 88.4±0.7 | 96.3±0.4 | 70.0±6.1 | 91.0±2.7 | 90.9±0.9 | 96.7±0.4 |
| Capsule | 72.9±0.0 | 88.1±0.0 | 57.7±7.3 | 67.8±2.9 | 72.3±6.8 | 83.6±6.9 | 64.3±3.0 | 72.8±7.0 | 77.3±8.8 | 89.0±2.7 | 65.2±2.5 | 72.8±7.0 | 82.3±8.9 | 91.2±1.6 |
| Carpet | 100.0±0.0 | 99.9±0.0 | 96.6±1.0 | 95.3±0.8 | 99.8±0.3 | 100.0±0.0 | 97.8±0.5 | 96.6±0.5 | 99.8±0.3 | 100.0±0.1 | 97.9±0.4 | 96.6±0.5 | 100.0±0.0 | 100.0±0.0 |
| Grid | 98.8±0.0 | 99.9±0.0 | 54.2±6.7 | 63.6±10.3 | 99.5±0.3 | 99.4±0.4 | 67.2±4.2 | 67.7±8.3 | 99.4±0.2 | 99.6±0.4 | 68.1±3.8 | 67.7±8.3 | 99.6±0.1 | 99.5±0.4 |
| Hazelnut | 93.9±0.0 | 96.9±0.0 | 88.3±2.6 | 88.3±2.7 | 97.5±1.4 | 99.3±0.4 | 90.8±0.8 | 93.2±3.8 | 98.3±0.7 | 99.3±0.3 | 91.9±1.2 | 93.2±3.8 | 98.4±0.4 | 99.3±0.5 |
| Leather | 100.0±0.0 | 99.9±0.0 | 97.5±0.7 | 97.3±0.7 | 99.9±0.0 | 99.9±0.0 | 97.5±0.9 | 97.9±0.7 | 99.9±0.0 | 100.0±0.0 | 98.5±0.2 | 97.9±0.7 | 100.0±0.0 | 100.0±0.0 |
| Metal nut | 97.1±0.0 | 98.3±0.0 | 53.0±3.8 | 73.4±2.9 | 98.7±0.8 | 99.8±0.3 | 54.8±3.8 | 77.7±8.5 | 99.4±0.2 | 99.9±0.2 | 60.7±5.2 | 77.7±8.5 | 99.5±0.2 | 100.0±0.0 |
| Pill | 79.1±0.0 | 77.6±0.0 | 61.3±3.8 | 81.9±2.8 | 91.2±2.1 | 96.1±0.4 | 59.1±6.4 | 82.9±2.9 | 92.3±0.7 | 96.2±0.2 | 54.9±2.7 | 82.9±2.9 | 92.8±1.0 | 96.4±0.4 |
| Screw | 83.3±0.0 | 77.3±0.0 | 55.0±2.5 | 44.4±4.6 | 86.4±0.9 | 72.5±5.5 | 54.0±4.4 | 49.0±3.8 | 86.0±2.1 | 73.5±5.1 | 50.0±4.1 | 49.0±3.8 | 87.9±1.2 | 82.0±4.5 |
| Tile | 100.0±0.0 | 99.1±0.0 | 92.2±2.2 | 99.0±0.9 | 99.9±0.0 | 99.9±0.2 | 93.3±1.1 | 98.5±1.0 | 99.9±0.2 | 99.9±0.1 | 93.1±0.6 | 98.5±1.0 | 99.9±0.1 | 100.0±0.0 |
| Toothbrush | 87.5±0.0 | 96.9±0.0 | 82.5±1.2 | 83.3±3.8 | 92.2±4.9 | 98.9±1.6 | 87.6±4.2 | 85.9±3.5 | 97.5±1.6 | 99.1±0.8 | 89.2±2.5 | 85.9±3.5 | 96.7±2.6 | 98.5±0.8 |
| Transistor | 88.0±0.0 | 86.7±0.0 | 73.3±6.0 | 78.1±6.9 | 83.4±3.8 | 90.9±1.4 | 72.8±6.3 | 90.0±4.3 | 85.3±1.7 | 91.1±0.8 | 82.4±6.5 | 90.0±4.3 | 85.7±2.5 | 92.9±1.0 |
| Wood | 99.4±0.0 | 96.7±0.0 | 96.1±1.2 | 97.8±0.3 | 99.9±0.1 | 98.9±0.3 | 96.9±0.5 | 98.3±0.6 | 99.9±0.1 | 98.6±0.2 | 97.0±0.2 | 98.3±0.6 | 99.9±0.3 | 98.8±0.1 |
| Zipper | 91.5±0.0 | 94.0±0.0 | 85.8±2.7 | 92.3±0.5 | 88.8±5.9 | 96.5±0.7 | 86.3±2.6 | 94.0±2.1 | 94.0±1.4 | 95.3±2.2 | 88.3±2.0 | 94.0±2.1 | 94.5±0.5 | 96.2±0.6 |
| **Mean** | 91.8±0.0 | **93.2±0.0** | 76.6±3.1 | 83.4±3.0 | 93.1±2.0 | **95.4±0.8** | 78.9±3.1 | 86.3±3.3 | 94.4±1.3 | **95.9±0.3** | 80.4±2.5 | 88.8±2.6 | 95.2±1.3 | **96.8±0.2** |

Table 18: AUPR (%) results on MVTec-AD classification.

| Category | k=0 | | k=1 | | | | k=2 | | | | k=4 | | | |
|---|---|---|---|---|---|---|---|---|---|---|---|---|---|---|
| | WinCLIP | CLIPFUSION | PaDiM | PatchCore | WinCLIP | CLIPFUSION | PaDiM | PatchCore | WinCLIP | CLIPFUSION | PaDiM | PatchCore | WinCLIP | CLIPFUSION |
| Bottle | 99.8±0.0 | 100.0±0.0 | 99.2±0.2 | 99.8±0.1 | 99.4±0.3 | 100.0±0.0 | 99.6±0.3 | 99.8±0.1 | 99.8±0.1 | 100.0±0.0 | 99.7±0.0 | 99.8±0.1 | 99.8±0.1 | 100.0±0.0 |
| Cable | 91.2±0.0 | 92.7±0.0 | 64.9±3.8 | 93.8±2.2 | 93.2±1.1 | 97.1±0.5 | 69.6±6.6 | 95.1±1.3 | 92.9±0.6 | 97.8±0.3 | 76.1±5.6 | 97.1±0.7 | 94.4±0.3 | 98.1±0.2 |
| Capsule | 91.5±0.0 | 97.2±0.0 | 86.9±2.2 | 89.4±2.0 | 91.6±2.7 | 95.9±2.2 | 88.4±0.8 | 91.0±2.9 | 93.3±3.6 | 97.4±0.8 | 87.8±0.8 | 94.9±1.1 | 95.1±3.3 | 98.0±0.4 |
| Carpet | 100.0±0.0 | 100.0±0.0 | 99.0±0.2 | 98.7±0.2 | 99.9±0.1 | 100.0±0.0 | 99.4±0.1 | 99.0±0.1 | 99.9±0.1 | 100.0±0.0 | 99.4±0.1 | 98.8±0.2 | 100.0±0.0 | 100.0±0.0 |
| Grid | 99.6±0.0 | 100.0±0.0 | 75.0±3.3 | 81.1±4.9 | 99.9±0.1 | 99.8±0.1 | 82.5±2.3 | 84.1±4.0 | 99.8±0.1 | 99.9±0.1 | 83.0±1.8 | 86.4±4.0 | 99.9±0.0 | 99.8±0.1 |
| Hazelnut | 96.9±0.0 | 98.3±0.0 | 93.3±1.7 | 92.9±2.2 | 98.6±0.7 | 99.6±0.2 | 94.1±0.5 | 96.0±2.0 | 99.1±0.4 | 99.6±0.1 | 94.8±0.6 | 97.0±1.2 | 99.1±0.2 | 99.6±0.3 |
| Leather | 100.0±0.0 | 100.0±0.0 | 99.2±0.2 | 99.1±0.2 | 100.0±0.0 | 100.0±0.0 | 99.2±0.3 | 99.3±0.2 | 100.0±0.0 | 100.0±0.0 | 99.6±0.1 | 99.6±0.1 | 100.0±0.0 | 100.0±0.0 |
| Metal nut | 99.3±0.0 | 99.6±0.0 | 82.0±2.7 | 91.0±1.1 | 99.7±0.2 | 99.9±0.1 | 82.2±1.4 | 92.3±4.0 | 99.9±0.0 | 100.0±0.0 | 85.5±1.7 | 97.0±2.6 | 99.9±0.1 | 100.0±0.0 |
| Pill | 95.7±0.0 | 94.8±0.0 | 88.3±1.3 | 96.5±0.6 | 98.3±0.5 | 99.3±0.1 | 87.9±2.6 | 96.6±0.7 | 98.6±0.1 | 99.3±0.0 | 87.0±1.2 | 96.9±0.4 | 98.5±0.2 | 99.4±0.1 |
| Screw | 93.1±0.0 | 89.3±0.0 | 78.1±1.0 | 71.4±2.3 | 94.2±0.6 | 87.5±4.7 | 77.3±1.3 | 72.9±3.4 | 94.1±1.5 | 87.4±3.2 | 75.7±2.8 | 71.8±1.9 | 94.9±0.8 | 93.0±2.3 |
| Tile | 100.0±0.0 | 99.6±0.0 | 97.2±0.7 | 99.6±0.3 | 100.0±0.0 | 100.0±0.1 | 97.6±0.4 | 99.4±0.4 | 100.0±0.1 | 100.0±0.0 | 97.6±0.2 | 99.6±0.1 | 100.0±0.0 | 100.0±0.0 |
| Toothbrush | 95.6±0.0 | 98.8±0.0 | 93.7±0.5 | 93.5±1.4 | 96.7±2.0 | 99.6±0.5 | 95.2±1.6 | 94.1±1.4 | 99.0±0.6 | 99.7±0.3 | 95.8±0.7 | 94.8±0.7 | 98.7±1.1 | 99.5±0.2 |
| Transistor | 87.1±0.0 | 84.7±0.0 | 66.2±7.5 | 77.7±5.5 | 79.0±4.0 | 89.8±1.5 | 69.0±6.5 | 89.3±3.9 | 80.7±2.3 | 89.9±0.7 | 77.6±8.4 | 84.5±9.0 | 80.7±3.2 | 92.3±1.3 |
| Wood | 99.8±0.0 | 99.0±0.0 | 98.8±0.3 | 99.3±0.1 | 100.0±0.0 | 99.7±0.1 | 99.0±0.1 | 99.5±0.2 | 100.0±0.0 | 99.6±0.1 | 99.1±0.0 | 99.5±0.2 | 99.9±0.1 | 99.7±0.0 |
| Zipper | 97.5±0.0 | 97.9±0.0 | 95.5±0.9 | 97.2±0.3 | 96.8±1.8 | 98.9±0.3 | 95.4±1.0 | 97.8±1.0 | 98.3±0.4 | 98.3±1.0 | 96.2±0.8 | 99.1±0.7 | 98.5±0.2 | 98.7±0.2 |
| **Mean** | 96.5±0.0 | **96.8±0.0** | 88.1±1.7 | 92.2±1.5 | 96.5±0.9 | **97.8±0.4** | 89.3±1.7 | 93.8±1.7 | 97.0±0.7 | **97.9±0.2** | 90.5±1.6 | 94.5±1.5 | 97.3±0.6 | **98.5±0.1** |

Table 19: AUROC (%) results on MVTec-AD segmentation.

| Category | k=0 | | k=1 | | | | k=2 | | | | k=4 | | | |
|---|---|---|---|---|---|---|---|---|---|---|---|---|---|---|
| | WinCLIP | CLIPFUSION | PaDiM | PatchCore | WinCLIP | CLIPFUSION | PaDiM | PatchCore | WinCLIP | CLIPFUSION | PaDiM | PatchCore | WinCLIP | CLIPFUSION |
| Bottle | 89.5±0.0 | 94.1±0.0 | 96.1±0.5 | 97.9±0.1 | 97.5±0.2 | 97.3±0.2 | 96.9±0.1 | 98.1±0.0 | 97.7±0.1 | 97.5±0.0 | 97.1±0.1 | 98.2±0.0 | 97.8±0.0 | 97.7±0.1 |
| Cable | 77.0±0.0 | 73.2±0.0 | 88.4±1.2 | 95.5±0.8 | 93.8±0.6 | 91.3±0.2 | 90.0±0.8 | 96.4±0.3 | 94.3±0.4 | 92.2±0.3 | 92.1±0.4 | 97.5±0.3 | 94.9±0.1 | 92.9±0.4 |
| Capsule | 86.9±0.0 | 94.6±0.0 | 94.5±0.6 | 95.6±0.4 | 94.6±0.8 | 97.9±0.4 | 95.2±0.5 | 96.5±0.4 | 96.4±0.3 | 98.4±0.2 | 96.2±0.4 | 96.8±0.6 | 96.2±0.5 | 98.5±0.1 |
| Carpet | 95.4±0.0 | 99.3±0.0 | 97.8±0.2 | 98.4±0.1 | 99.4±0.0 | 99.4±0.1 | 98.2±0.0 | 98.5±0.1 | 99.3±0.0 | 99.5±0.0 | 98.4±0.0 | 98.6±0.1 | 99.3±0.0 | 99.5±0.0 |
| Grid | 82.2±0.0 | 98.9±0.0 | 70.2±2.8 | 58.8±4.9 | 96.8±1.0 | 98.8±0.2 | 70.8±2.0 | 62.6±3.2 | 97.7±0.8 | 98.9±0.4 | 77.0±1.8 | 69.4±1.3 | 98.0±0.2 | 99.2±0.1 |
| Hazelnut | 94.3±0.0 | 96.7±0.0 | 95.4±0.7 | 95.8±0.6 | 98.5±0.2 | 98.2±0.1 | 96.8±0.3 | 96.3±0.6 | 98.7±0.1 | 98.2±0.1 | 97.2±0.2 | 97.6±0.1 | 98.8±0.0 | 98.7±0.1 |
| Leather | 96.7±0.0 | 99.2±0.0 | 98.5±0.1 | 98.8±0.2 | 99.3±0.0 | 99.3±0.0 | 98.7±0.1 | 99.0±0.1 | 99.3±0.0 | 99.3±0.0 | 98.8±0.0 | 99.1±0.0 | 99.3±0.0 | 99.4±0.0 |
| Metal nut | 61.0±0.0 | 74.2±0.0 | 74.6±1.1 | 89.3±1.4 | 90.0±0.6 | 86.2±1.2 | 80.3±2.1 | 94.6±1.4 | 91.4±0.4 | 89.7±0.9 | 82.7±3.9 | 95.9±1.8 | 92.9±0.4 | 91.3±0.8 |
| Pill | 80.0±0.0 | 83.0±0.0 | 84.8±1.0 | 93.1±1.1 | 96.4±0.3 | 95.5±0.2 | 87.3±0.7 | 94.2±0.3 | 97.0±0.2 | 95.6±0.2 | 88.9±0.5 | 94.8±0.4 | 97.1±0.0 | 95.8±0.2 |
| Screw | 89.6±0.0 | 97.2±0.0 | 83.3±0.7 | 89.6±0.5 | 94.5±0.4 | 95.6±0.4 | 89.8±0.8 | 90.0±0.7 | 95.2±0.3 | 95.6±0.6 | 90.8±0.2 | 91.3±1.0 | 96.0±0.5 | 96.3±0.5 |
| Tile | 77.6±0.0 | 96.2±0.0 | 84.1±1.1 | 94.1±0.5 | 96.3±0.2 | 97.7±0.1 | 87.7±0.2 | 94.4±0.2 | 96.5±0.1 | 97.8±0.1 | 88.9±0.3 | 94.6±0.1 | 96.6±0.1 | 97.8±0.1 |
| Toothbrush | 86.9±0.0 | 93.2±0.0 | 97.3±0.3 | 97.3±0.4 | 97.8±0.1 | 99.1±0.0 | 97.7±0.3 | 97.5±0.2 | 98.1±0.1 | 99.2±0.1 | 98.4±0.2 | 98.4±0.0 | 98.4±0.5 | 99.2±0.1 |
| Transistor | 74.7±0.0 | 69.2±0.0 | 90.2±2.8 | 84.9±2.7 | 85.0±1.8 | 87.3±2.6 | 92.3±2.1 | 89.6±0.9 | 88.3±1.0 | 88.6±0.8 | 94.0±2.7 | 90.7±1.4 | 88.5±1.2 | 90.3±0.6 |
| Wood | 93.4±0.0 | 96.0±0.0 | 90.7±0.4 | 92.7±0.9 | 94.6±1.0 | 95.7±1.2 | 91.9±0.1 | 93.2±0.7 | 95.3±0.4 | 96.4±0.3 | 92.2±0.1 | 93.5±0.3 | 95.4±0.2 | 96.4±0.4 |
| Zipper | 91.6±0.0 | 98.2±0.0 | 93.9±0.8 | 97.4±0.4 | 93.9±0.8 | 98.4±0.3 | 95.4±0.3 | 98.0±0.1 | 94.1±0.7 | 98.3±0.1 | 96.1±0.2 | 98.1±0.1 | 94.2±0.4 | 98.5±0.1 |
| **Mean** | 85.1±0.0 | **90.9±0.0** | 89.3±0.9 | 92.0±1.0 | 95.2±0.5 | **95.8±0.1** | 91.3±0.7 | 93.3±0.6 | 96.0±0.3 | **96.3±0.1** | 92.6±0.7 | 94.3±0.5 | 96.2±0.3 | **96.8±0.1** |

Table 20: AUPRO (%) results on MVTec-AD segmentation.

| Category | k=0 | | k=1 | | | | k=2 | | | | k=4 | | | |
|---|---|---|---|---|---|---|---|---|---|---|---|---|---|---|
| | WinCLIP | CLIPFUSION | PaDiM | PatchCore | WinCLIP | CLIPFUSION | PaDiM | PatchCore | WinCLIP | CLIPFUSION | PaDiM | PatchCore | WinCLIP | CLIPFUSION |
| Bottle | 76.4±0.0 | 88.4±0.0 | 89.8±0.8 | 93.5±0.3 | 91.2±0.4 | 93.7±0.4 | 91.7±0.2 | 93.9±0.3 | 91.8±0.3 | 94.1±0.1 | 92.2±0.2 | 94.0±0.2 | 91.6±0.2 | 94.5±0.3 |
| Cable | 42.9±0.0 | 71.5±0.0 | 59.1±3.2 | 84.7±1.0 | 72.5±2.3 | 79.7±0.8 | 66.5±2.8 | 88.5±0.9 | 74.7±2.3 | 82.0±1.0 | 74.2±1.8 | 91.7±0.6 | 77.0±1.1 | 84.4±1.0 |
| Capsule | 62.1±0.0 | 88.5±0.0 | 80.0±2.0 | 83.9±0.9 | 85.6±2.7 | 92.5±1.1 | 82.3±2.1 | 86.6±1.0 | 90.6±0.6 | 94.1±0.2 | 85.7±1.3 | 87.8±1.9 | 90.1±1.5 | 94.7±0.5 |
| Carpet | 84.1±0.0 | 98.2±0.0 | 92.9±0.3 | 93.3±0.3 | 97.4±0.4 | 98.3±0.1 | 93.9±0.2 | 93.7±0.4 | 97.3±0.3 | 98.4±0.1 | 94.4±0.2 | 93.9±0.4 | 97.0±0.2 | 98.5±0.1 |
| Grid | 57.0±0.0 | 95.7±0.0 | 41.2±4.6 | 21.7±9.5 | 90.5±2.7 | 96.1±1.1 | 45.1±3.6 | 23.7±3.8 | 92.8±2.5 | 97.1±0.9 | 55.5±3.4 | 30.4±4.6 | 93.6±0.6 | 96.9±0.7 |
| Hazelnut | 81.6±0.0 | 93.8±0.0 | 85.7±1.9 | 88.3±1.3 | 93.7±0.9 | 94.3±0.3 | 89.4±0.9 | 89.8±1.3 | 94.2±0.3 | 94.7±0.5 | 90.4±0.7 | 92.0±0.3 | 94.2±0.3 | 95.6±0.6 |
| Leather | 91.1±0.0 | 97.7±0.0 | 95.6±0.2 | 95.2±1.0 | 98.6±0.0 | 97.8±0.1 | 96.2±0.2 | 95.9±0.3 | 98.3±0.4 | 97.7±0.1 | 96.3±0.1 | 96.4±0.1 | 98.0±0.4 | 97.6±0.1 |
| Metal nut | 31.8±0.0 | 74.1±0.0 | 38.1±1.6 | 66.7±2.9 | 84.7±1.1 | 80.1±1.6 | 48.2±5.0 | 79.6±4.2 | 86.7±0.8 | 85.8±1.6 | 54.0±8.8 | 83.8±5.5 | 89.4±0.1 | 87.6±1.1 |
| Pill | 65.0±0.0 | 87.6±0.0 | 78.9±0.6 | 89.5±1.6 | 93.5±0.2 | 93.9±0.3 | 84.3±0.4 | 91.6±0.5 | 94.5±0.2 | 94.2±0.2 | 86.6±0.4 | 92.5±0.4 | 94.6±0.3 | 94.5±0.1 |
| Screw | 68.5±0.0 | 89.5±0.0 | 51.6±1.7 | 68.1±1.3 | 82.3±1.1 | 84.0±0.8 | 69.5±2.1 | 69.0±2.1 | 84.1±0.5 | 83.6±1.5 | 72.3±0.8 | 72.4±3.1 | 86.3±1.8 | 85.9±1.1 |
| Tile | 51.2±0.0 | 91.9±0.0 | 66.7±1.5 | 82.5±1.1 | 89.4±0.4 | 94.5±0.2 | 71.9±0.5 | 82.5±0.5 | 89.6±0.4 | 94.7±0.2 | 73.6±0.9 | 83.0±0.1 | 89.9±0.3 | 94.8±0.1 |
| Toothbrush | 67.7±0.0 | 90.6±0.0 | 82.1±1.5 | 79.0±2.4 | 85.3±1.0 | 91.6±0.9 | 83.3±2.6 | 81.0±0.7 | 84.7±1.4 | 92.1±0.3 | 87.1±1.7 | 85.5±3.0 | 86.0±3.3 | 92.7±0.4 |
| Transistor | 43.4±0.0 | 52.5±0.0 | 70.3±7.0 | 70.9±4.6 | 65.0±1.8 | 68.8±3.0 | 76.5±5.5 | 78.8±1.5 | 68.6±1.1 | 71.3±0.6 | 82.2±7.4 | 79.5±2.8 | 69.6±1.1 | 73.1±1.3 |
| Wood | 74.1±0.0 | 93.1±0.0 | 86.5±0.6 | 87.1±1.0 | 91.0±0.6 | 95.0±0.8 | 88.0±0.2 | 86.8±1.4 | 91.8±0.6 | 95.3±0.2 | 88.4±0.2 | 87.7±0.4 | 91.7±0.3 | 95.3±0.4 |
| Zipper | 71.7±0.0 | 94.0±0.0 | 81.7±2.0 | 91.2±1.1 | 86.0±1.7 | 95.4±0.6 | 85.6±0.7 | 92.8±0.4 | 86.4±1.6 | 95.7±0.3 | 87.2±0.8 | 93.4±0.2 | 86.9±0.7 | 96.0±0.3 |
| **Mean** | 64.6±0.0 | **87.1±0.0** | 73.3±2.0 | 79.7±2.0 | 87.1±1.2 | **90.4±0.1** | 78.2±1.8 | 82.3±1.3 | 88.4±0.9 | **91.4±0.2** | 81.3±1.9 | 84.3±1.6 | 89.0±0.8 | **92.1±0.2** |

Table 21: AUROC (%) results on VisA classification.

| Category | k=0 | | k=1 | | | | k=2 | | | | k=4 | | | |
|---|---|---|---|---|---|---|---|---|---|---|---|---|---|---|
| | WinCLIP | CLIPFUSION | PaDiM | PatchCore | WinCLIP | CLIPFUSION | PaDiM | PatchCore | WinCLIP | CLIPFUSION | PaDiM | PatchCore | WinCLIP | CLIPFUSION |
| Candle | 95.4±0.0 | 94.3±0.0 | 70.8±4.1 | 85.1±1.4 | 93.4±1.4 | 92.6±0.7 | 75.8±2.1 | 85.3±1.5 | 94.8±1.0 | 93.4±1.8 | 77.5±1.6 | 87.8±0.8 | 95.1±0.3 | 94.0±0.9 |
| Capsules | 85.0±0.0 | 80.3±0.0 | 51.0±7.8 | 60.0±7.6 | 85.0±3.1 | 83.0±1.2 | 51.7±4.6 | 57.8±5.4 | 84.9±0.8 | 83.1±0.9 | 52.7±3.4 | 63.4±5.4 | 86.8±1.7 | 83.0±0.8 |
| Cashew | 92.1±0.0 | 81.7±0.0 | 62.3±9.9 | 89.5±4.4 | 94.0±0.4 | 89.4±1.7 | 74.6±3.6 | 93.6±0.6 | 94.3±0.5 | 89.3±1.0 | 77.7±3.2 | 93.0±1.5 | 95.2±0.8 | 90.8±1.4 |
| Chewinggum | 96.5±0.0 | 96.8±0.0 | 69.9±4.9 | 97.3±0.3 | 97.6±0.8 | 97.4±0.3 | 82.7±2.1 | 97.8±0.6 | 97.3±0.8 | 97.6±0.3 | 83.5±3.7 | 98.3±0.3 | 97.7±0.3 | 97.7±0.3 |
| Fryum | 80.3±0.0 | 88.4±0.0 | 58.3±5.9 | 75.0±4.8 | 88.5±1.9 | 92.2±1.0 | 69.2±9.0 | 83.4±2.4 | 90.5±0.4 | 93.1±1.2 | 71.2±5.9 | 88.6±1.3 | 90.8±0.5 | 93.7±0.4 |
| Macaroni1 | 76.2±0.0 | 79.3±0.0 | 62.1±4.6 | 68.0±3.4 | 82.9±1.5 | 86.8±0.7 | 62.2±5.0 | 75.6±4.6 | 83.3±1.9 | 87.9±1.1 | 65.9±3.9 | 82.9±2.7 | 85.2±0.9 | 88.9±1.4 |
| Macaroni2 | 63.7±0.0 | 61.8±0.0 | 47.5±5.9 | 55.6±4.6 | 70.2±0.9 | 72.2±2.1 | 50.8±2.9 | 57.3±5.6 | 71.8±2.0 | 70.8±2.2 | 55.0±2.9 | 61.7±1.8 | 70.9±2.2 | 72.7±1.8 |
| PCB1 | 73.6±0.0 | 83.7±0.0 | 76.2±1.2 | 78.9±1.1 | 75.6±23.0 | 91.3±1.3 | 62.4±10.8 | 71.5±20.0 | 76.7±5.2 | 89.7±4.0 | 82.6±1.5 | 84.7±6.7 | 88.3±1.7 | 89.7±2.8 |
| PCB2 | 51.2±0.0 | 61.6±0.0 | 61.2±2.0 | 81.5±0.8 | 62.2±3.9 | 69.7±5.3 | 66.8±2.0 | 84.3±1.7 | 62.6±3.7 | 75.9±3.3 | 73.5±2.4 | 84.3±1.0 | 67.5±2.6 | 76.4±1.4 |
| PCB3 | 73.4±0.0 | 67.8±0.0 | 51.4±12.2 | 82.7±2.3 | 74.1±1.1 | 76.6±4.5 | 67.3±3.8 | 84.8±1.2 | 78.8±1.9 | 79.2±2.7 | 65.9±1.9 | 87.0±1.1 | 83.3±1.7 | 78.4±1.2 |
| PCB4 | 79.6±0.0 | 76.9±0.0 | 76.1±3.6 | 93.9±2.8 | 85.2±8.9 | 91.4±1.9 | 69.3±13.7 | 94.3±3.2 | 82.3±9.9 | 87.8±5.0 | 85.4±2.0 | 95.6±1.6 | 87.6±8.0 | 94.1±1.6 |
| Pipe fryum | 69.7±0.0 | 81.5±0.0 | 66.7±2.2 | 90.7±1.7 | 97.2±1.1 | 95.8±1.1 | 75.3±1.8 | 93.5±1.3 | 98.0±0.6 | 96.6±0.6 | 82.9±2.2 | 96.4±0.7 | 98.5±0.4 | 97.0±0.2 |
| **Mean** | 78.1±0.0 | 79.5±0.0 | 62.8±5.4 | 79.9±2.9 | 83.8±4.0 | **86.5±0.1** | 67.4±5.1 | 81.6±4.0 | 84.6±2.4 | **87.0±0.5** | 72.8±2.9 | 85.3±2.1 | 87.3±1.8 | **88.0±0.2** |

Table 22: AUPR (%) results on VisA classification.

| Category | k=0 | | k=1 | | | | k=2 | | | | k=4 | | | |
|---|---|---|---|---|---|---|---|---|---|---|---|---|---|---|
| | WinCLIP | CLIPFUSION | PaDiM | PatchCore | WinCLIP | CLIPFUSION | PaDiM | PatchCore | WinCLIP | CLIPFUSION | PaDiM | PatchCore | WinCLIP | CLIPFUSION |
| Candle | 95.8±0.0 | 94.9±0.0 | 69.2±3.9 | 86.6±2.3 | 93.6±1.5 | 92.9±0.6 | 72.8±1.0 | 86.8±1.7 | 95.1±1.1 | 93.9±1.8 | 72.5±1.1 | 88.9±1.1 | 95.3±0.4 | 94.4±1.0 |
| Capsules | 90.9±0.0 | 88.8±0.0 | 63.4±5.7 | 72.3±5.3 | 89.9±2.5 | 90.4±0.7 | 63.4±2.0 | 73.6±4.7 | 88.9±0.7 | 90.6±0.4 | 63.0±2.3 | 78.4±3.1 | 91.5±1.4 | 90.7±0.3 |
| Cashew | 96.4±0.0 | 91.9±0.0 | 78.2±5.7 | 94.6±2.0 | 97.2±0.2 | 95.3±0.8 | 86.1±2.2 | 96.9±0.3 | 97.3±0.2 | 95.2±0.4 | 88.4±2.0 | 96.5±0.7 | 97.7±0.4 | 95.8±0.6 |
| Chewinggum | 98.6±0.0 | 98.7±0.0 | 79.8±3.6 | 98.9±0.1 | 99.0±0.3 | 98.9±0.1 | 89.5±1.9 | 99.1±0.2 | 98.9±0.3 | 99.0±0.1 | 88.5±3.2 | 99.3±0.1 | 99.0±0.1 | 99.0±0.1 |
| Fryum | 90.1±0.0 | 94.6±0.0 | 74.5±2.9 | 87.6±2.4 | 94.7±1.0 | 96.6±0.4 | 81.0±5.4 | 92.1±1.3 | 95.8±0.2 | 96.9±0.5 | 81.5±3.0 | 95.0±0.6 | 96.0±0.3 | 97.1±0.1 |
| Macaroni1 | 75.8±0.0 | 81.1±0.0 | 60.4±2.9 | 67.8±3.4 | 84.9±1.2 | 88.4±0.9 | 63.1±4.3 | 74.9±5.2 | 84.7±1.5 | 88.9±0.9 | 64.9±2.1 | 82.1±3.5 | 86.5±0.6 | 89.8±1.3 |
| Macaroni2 | 60.3±0.0 | 59.5±0.0 | 51.7±5.0 | 54.9±3.2 | 68.4±1.8 | 71.0±2.4 | 52.7±1.5 | 57.2±2.6 | 70.4±1.8 | 68.3±4.9 | 54.9±2.5 | 60.2±3.0 | 69.6±2.8 | 70.9±3.3 |
| PCB1 | 78.4±0.0 | 87.0±0.0 | 68.6±2.4 | 72.1±2.5 | 76.5±19.0 | 91.2±1.4 | 60.4±7.7 | 72.6±16.4 | 78.3±4.3 | 89.2±4.2 | 77.4±2.9 | 81.0±9.2 | 87.7±1.7 | 89.3±3.5 |
| PCB2 | 49.2±0.0 | 66.8±0.0 | 63.3±1.2 | 84.4±0.4 | 64.9±3.3 | 69.8±5.1 | 68.9±2.6 | 86.6±1.1 | 65.8±4.0 | 75.7±3.2 | 75.0±1.7 | 86.2±1.0 | 71.3±3.4 | 75.5±2.0 |
| PCB3 | 76.5±0.0 | 70.3±0.0 | 52.3±10.8 | 84.6±1.5 | 73.5±1.6 | 79.5±3.3 | 65.2±3.8 | 86.1±0.5 | 80.9±1.6 | 80.9±2.9 | 64.5±2.4 | 88.3±1.1 | 84.8±1.8 | 81.0±1.4 |
| PCB4 | 77.7±0.0 | 78.5±0.0 | 74.7±2.6 | 92.8±3.1 | 78.5±15.5 | 88.8±3.1 | 67.6±11.9 | 93.2±3.4 | 72.5±16.2 | 84.8±5.6 | 84.0±2.0 | 94.9±1.2 | 85.6±8.9 | 91.9±2.1 |
| Pipe fryum | 82.3±0.0 | 90.5±0.0 | 79.2±1.5 | 95.4±0.6 | 98.6±0.5 | 98.0±0.6 | 84.5±1.7 | 96.8±0.7 | 99.0±0.3 | 98.4±0.3 | 89.8±1.7 | 98.3±0.3 | 99.2±0.2 | 98.6±0.1 |
| **Mean** | 81.2±0.0 | **83.6±0.0** | 68.3±4.0 | 82.8±2.3 | 85.1±4.0 | **88.4±0.2** | 71.6±3.8 | 84.8±3.2 | 85.8±2.7 | **88.5±0.5** | 75.6±2.2 | 87.5±2.1 | 88.8±1.8 | **89.5±0.1** |

Table 23: AUROC (%) results on VisA segmentation.

| Category | k=0 | | k=1 | | | | k=2 | | | | k=4 | | | |
|---|---|---|---|---|---|---|---|---|---|---|---|---|---|---|
| | WinCLIP | CLIPFUSION | PaDiM | PatchCore | WinCLIP | CLIPFUSION | PaDiM | PatchCore | WinCLIP | CLIPFUSION | PaDiM | PatchCore | WinCLIP | CLIPFUSION |
| Candle | 88.9±0.0 | 97.5±0.0 | 91.7±2.2 | 97.2±0.2 | 97.4±0.2 | 98.4±0.2 | 94.9±0.8 | 97.7±0.3 | 97.7±0.1 | 98.5±0.1 | 95.4±0.2 | 97.9±0.1 | 97.8±0.2 | 98.6±0.1 |
| Capsules | 81.6±0.0 | 96.1±0.0 | 70.9±1.1 | 93.2±0.9 | 96.4±0.6 | 97.8±0.3 | 75.7±1.7 | 94.0±0.2 | 96.8±0.3 | 97.9±0.5 | 79.1±0.7 | 94.8±0.5 | 97.1±0.2 | 97.9±0.1 |
| Cashew | 84.7±0.0 | 72.5±0.0 | 95.5±0.6 | 98.1±0.1 | 98.5±0.2 | 98.2±0.3 | 96.4±0.4 | 98.2±0.2 | 98.5±0.1 | 98.5±0.1 | 97.2±0.3 | 98.3±0.2 | 98.7±0.0 | 98.0±0.5 |
| Chewinggum | 93.3±0.0 | 99.1±0.0 | 90.1±0.4 | 96.9±0.3 | 98.6±0.1 | 99.0±0.1 | 93.1±0.7 | 96.6±0.1 | 98.6±0.1 | 98.9±0.1 | 94.4±0.5 | 96.8±0.1 | 98.5±0.1 | 96.5±0.4 |
| Fryum | 88.5±0.0 | 92.5±0.0 | 93.3±0.6 | 93.3±0.5 | 96.4±0.3 | 95.2±0.3 | 94.1±0.6 | 94.0±0.3 | 97.0±0.2 | 96.1±0.3 | 95.0±0.4 | 94.2±0.2 | 97.1±0.1 | 96.5±0.4 |
| Macaroni1 | 70.9±0.0 | 98.6±0.0 | 89.4±0.9 | 95.2±0.4 | 96.4±0.6 | 99.1±0.1 | 91.7±0.3 | 96.0±1.3 | 96.5±0.7 | 99.2±0.1 | 93.5±0.5 | 97.0±0.3 | 97.0±0.2 | 99.3±0.1 |
| Macaroni2 | 59.3±0.0 | 96.9±0.0 | 86.4±1.1 | 89.1±1.6 | 96.8±0.4 | 97.4±0.3 | 90.1±0.8 | 90.2±1.9 | 96.8±0.6 | 97.5±0.1 | 90.2±0.3 | 93.9±0.3 | 97.3±0.3 | 97.9±0.2 |
| PCB1 | 61.2±0.0 | 91.6±0.0 | 89.9±0.3 | 96.1±1.5 | 96.6±0.6 | 95.5±0.2 | 90.6±0.6 | 97.6±0.9 | 97.0±0.9 | 96.2±0.2 | 93.2±1.5 | 98.1±1.0 | 98.1±0.9 | 97.4±0.9 |
| PCB2 | 71.6±0.0 | 91.1±0.0 | 90.9±1.4 | 95.4±0.2 | 93.0±0.4 | 96.1±0.3 | 93.9±0.9 | 96.0±0.3 | 93.9±0.2 | 96.7±0.1 | 93.7±1.0 | 96.6±0.2 | 94.6±0.4 | 97.3±0.3 |
| PCB3 | 85.3±0.0 | 88.4±0.0 | 93.9±0.3 | 96.2±0.3 | 94.3±0.3 | 96.2±0.2 | 95.1±0.5 | 97.1±0.1 | 95.1±0.2 | 96.9±0.2 | 95.7±0.1 | 97.4±0.2 | 95.8±0.1 | 97.3±0.2 |
| PCB4 | 94.4±0.0 | 93.5±0.0 | 89.6±0.6 | 95.6±0.6 | 94.0±0.9 | 95.8±0.5 | 90.7±0.9 | 96.2±0.4 | 95.6±0.3 | 96.7±0.2 | 92.1±0.5 | 97.0±0.2 | 96.1±0.3 | 97.3±0.2 |
| Pipe fryum | 75.4±0.0 | 87.6±0.0 | 97.2±0.6 | 98.8±0.2 | 98.3±0.2 | 99.4±0.1 | 98.1±0.4 | 99.1±0.1 | 98.5±0.2 | 99.4±0.1 | 98.5±0.1 | 99.1±0.0 | 98.7±0.1 | 99.5±0.0 |
| **Mean** | 79.6±0.0 | **92.1±0.0** | 89.9±0.8 | 95.4±0.6 | 96.4±0.4 | **97.3±0.1** | 92.0±0.7 | 96.1±0.5 | 96.8±0.3 | **97.7±0.1** | 93.2±0.5 | 96.8±0.3 | 97.2±0.2 | **98.0±0.1** |

Table 24: AUPRO (%) results on VisA segmentation.

| Category | k=0 | | k=1 | | | | k=2 | | | | k=4 | | | |
|---|---|---|---|---|---|---|---|---|---|---|---|---|---|---|
| | WinCLIP | CLIPFUSION | PaDiM | PatchCore | WinCLIP | CLIPFUSION | PaDiM | PatchCore | WinCLIP | CLIPFUSION | PaDiM | PatchCore | WinCLIP | CLIPFUSION |
| Candle | 83.5±0.0 | 93.3±0.0 | 81.5±5.3 | 92.6±0.4 | 94.0±0.4 | 94.5±0.7 | 87.3±1.2 | 93.4±0.6 | 94.2±0.2 | 94.8±0.3 | 88.3±0.7 | 94.1±0.4 | 94.4±0.2 | 95.3±0.3 |
| Capsules | 35.3±0.0 | 83.7±0.0 | 30.6±1.1 | 66.6±4.5 | 73.6±3.5 | 87.7±3.1 | 38.4±3.7 | 67.9±2.3 | 75.9±1.9 | 88.0±2.6 | 43.3±2.0 | 69.0±3.2 | 77.0±1.4 | 87.2±0.7 |
| Cashew | 76.4±0.0 | 92.3±0.0 | 73.4±2.1 | 90.8±0.2 | 91.1±0.8 | 91.7±1.4 | 78.4±2.7 | 91.4±1.0 | 90.4±0.6 | 90.1±1.2 | 81.2±2.8 | 92.1±0.3 | 91.3±0.9 | 89.6±1.5 |
| Chewinggum | 70.4±0.0 | 89.1±0.0 | 58.1±0.6 | 78.2±1.3 | 91.0±0.5 | 85.9±1.2 | 63.7±2.4 | 78.0±0.4 | 90.9±0.7 | 85.5±1.0 | 67.2±1.8 | 79.3±0.8 | 91.0±0.4 | 84.5±0.7 |
| Fryum | 77.4±0.0 | 88.3±0.0 | 71.1±1.6 | 78.7±2.3 | 89.1±1.0 | 83.2±2.1 | 71.2±0.8 | 81.4±2.8 | 89.3±0.2 | 84.1±1.7 | 73.2±1.3 | 81.0±1.2 | 89.7±0.5 | 85.3±1.4 |
| Macaroni1 | 34.3±0.0 | 87.7±0.0 | 62.2±4.4 | 83.4±1.3 | 84.6±2.3 | 90.0±0.5 | 71.8±2.4 | 86.2±4.6 | 85.2±1.4 | 91.2±0.8 | 76.6±2.1 | 89.6±0.7 | 86.8±0.8 | 91.8±0.7 |
| Macaroni2 | 21.4±0.0 | 75.5±0.0 | 54.9±3.6 | 66.0±3.0 | 89.3±2.4 | 80.3±2.0 | 65.6±3.4 | 67.2±6.5 | 88.6±1.7 | 79.2±1.5 | 65.9±1.5 | 78.3±0.9 | 90.5±1.3 | 83.0±1.8 |
| PCB1 | 26.3±0.0 | 73.5±0.0 | 63.9±1.8 | 79.0±10.7 | 82.5±6.0 | 86.3±0.7 | 68.4±4.1 | 86.1±1.7 | 83.8±5.0 | 86.1±2.9 | 70.2±3.3 | 88.1±2.6 | 87.9±2.1 | 84.9±2.8 |
| PCB2 | 37.2±0.0 | 73.5±0.0 | 64.4±3.8 | 80.9±0.5 | 73.6±1.5 | 80.5±1.8 | 72.9±3.4 | 82.9±1.8 | 76.2±0.9 | 82.8±0.9 | 71.9±2.6 | 83.7±1.0 | 78.0±1.3 | 84.0±1.9 |
| PCB3 | 56.1±0.0 | 66.2±0.0 | 69.0±1.2 | 78.1±2.0 | 79.5±2.5 | 82.7±1.9 | 74.0±2.3 | 82.3±1.1 | 82.3±1.8 | 83.1±1.6 | 77.2±0.8 | 84.4±1.9 | 84.2±1.0 | 81.7±3.1 |
| PCB4 | 80.4±0.0 | 81.2±0.0 | 59.1±1.8 | 77.9±3.1 | 76.6±4.1 | 84.5±1.8 | 62.6±3.6 | 79.5±4.8 | 81.7±1.2 | 86.1±1.2 | 67.9±2.6 | 83.5±2.5 | 84.2±0.7 | 88.5±1.0 |
| Pipe fryum | 82.3±0.0 | 95.2±0.0 | 83.9±0.8 | 93.6±0.5 | 96.1±0.6 | 96.6±0.3 | 86.9±0.9 | 94.5±0.4 | 96.2±0.6 | 97.2±0.2 | 88.7±1.3 | 95.0±0.5 | 96.6±0.2 | 97.0±0.2 |
| **Mean** | 56.8±0.0 | **83.3±0.0** | 64.3±2.4 | 80.5±2.5 | 85.1±2.1 | **87.0±0.3** | 70.1±2.6 | 82.6±2.3 | 86.2±1.4 | **87.3±0.2** | 72.6±1.9 | 84.9±1.4 | 87.6±0.9 | **87.7±0.5** |

## A.11   Detailed qualitative results

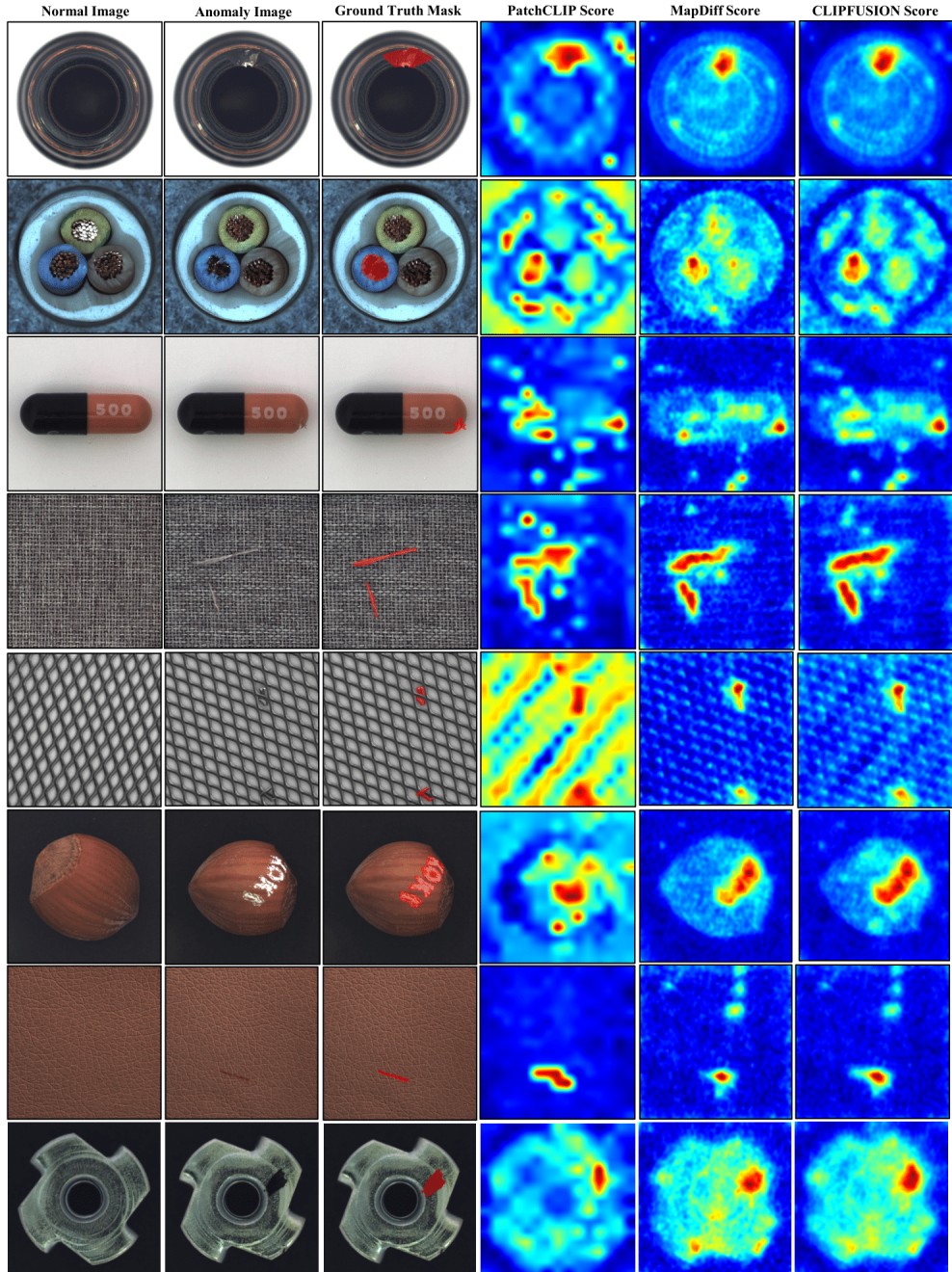

Figure 6: Qualitative results for 0-shot CLIPFUSION with images for PatchCLIP and MapDiff anomaly score maps on MVTec-AD.

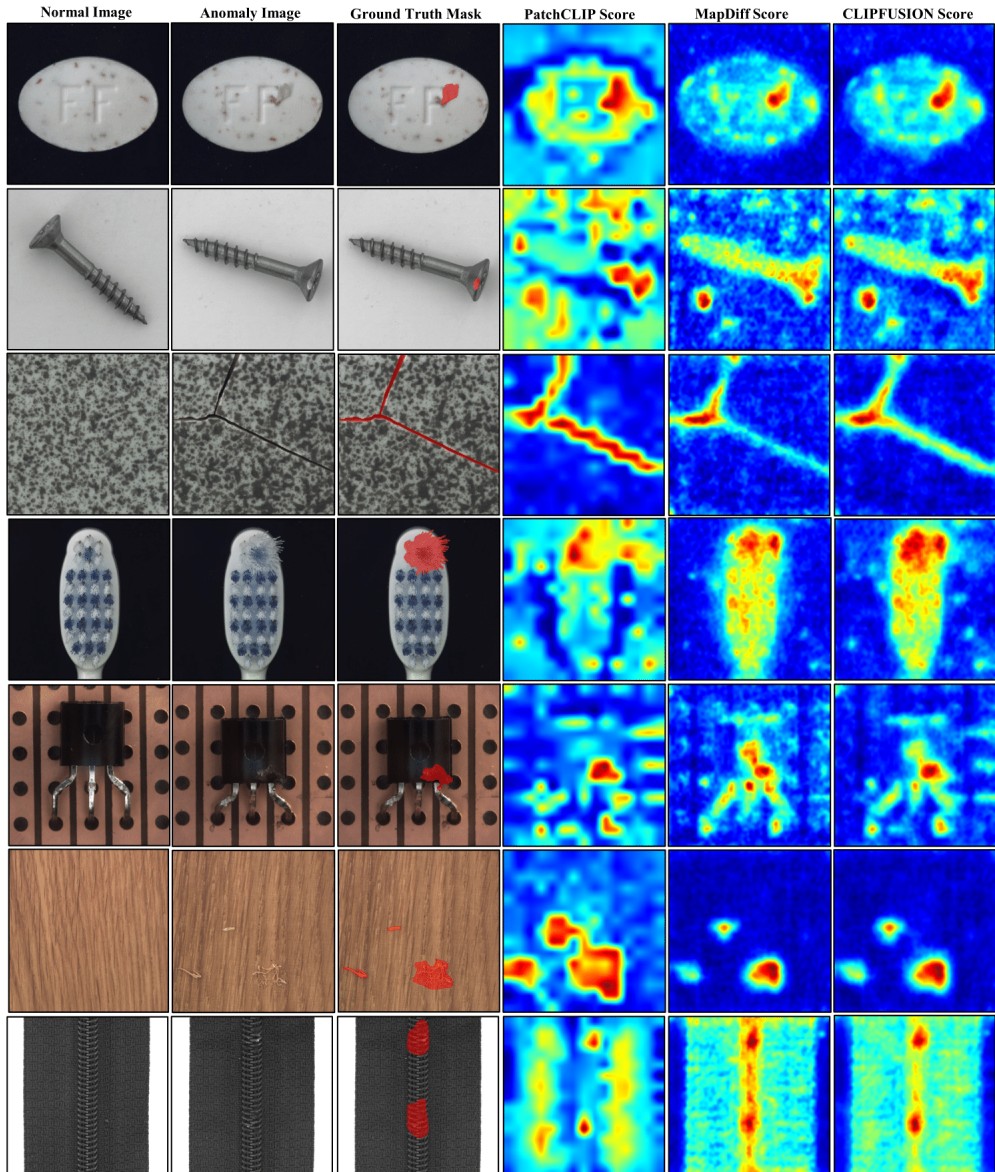

Figure 7: Qualitative results for 0-shot CLIPFUSION with images for PatchCLIP and MapDiff anomaly score maps on MVTec-AD.

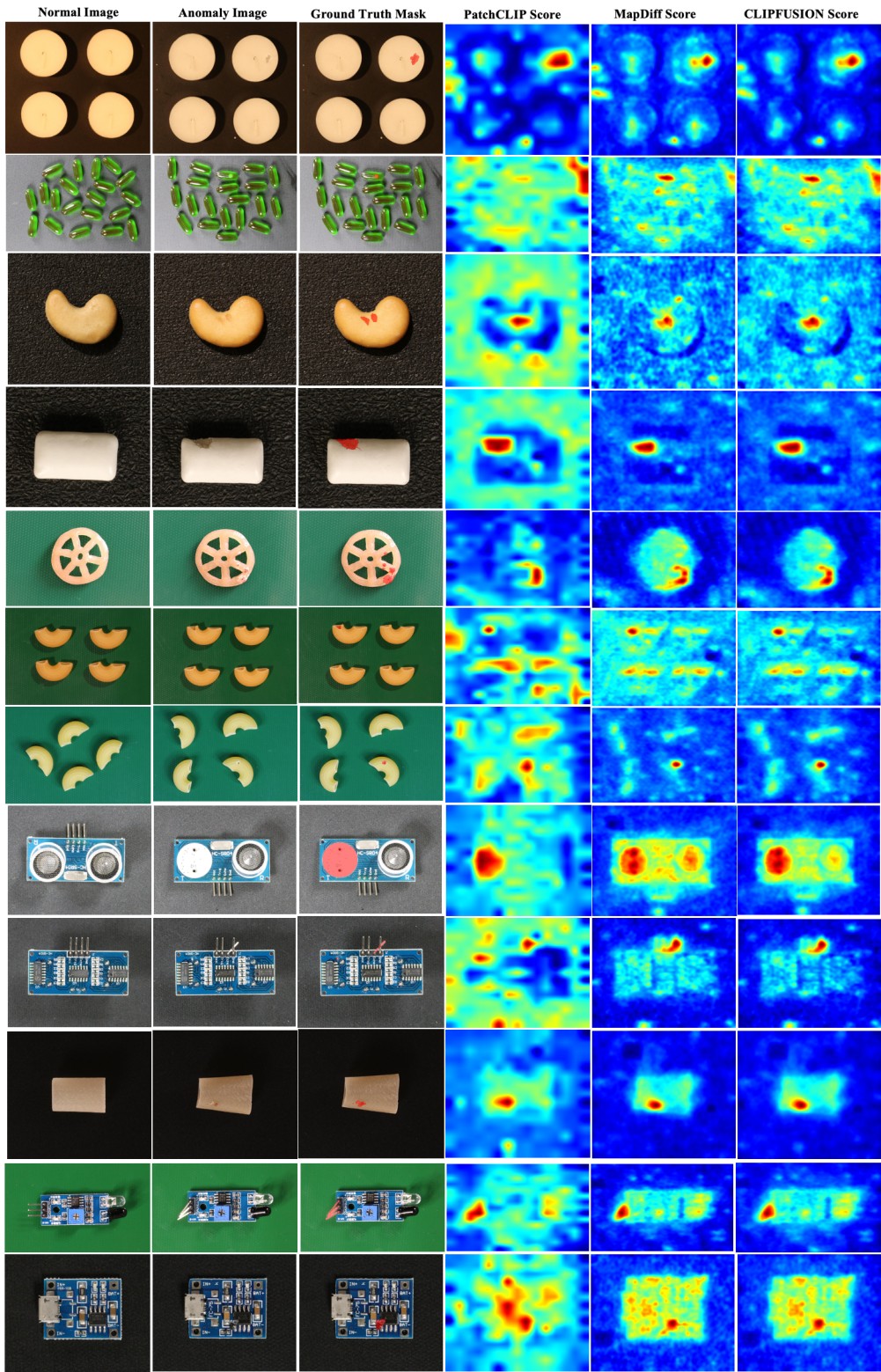

Figure 8: Qualitative results for 0-shot CLIPFUSION with images for PatchCLIP and MapDiff anomaly score maps on VisA.

