# OpenReview forum: "CLIP Meets Diffusion: A Synergistic Approach to Anomaly Detection"
_TMLR — Accepted by TMLR_

### Review · Reviewer_akeK · 2025-08-21

**Summary Of Contributions:**

The method proposes an anomaly detection method that fuses the feature maps from a CLIP and a diffusion model.  Then, it uses these fused features to compute the anomaly score.  The method is compared against different methods for 0 up to 4 shot classification in 2 datasets, and some ablation studies are included as well.

**Strengths:**
- The idea of mixing local and global features to improve the prediction of anomalies is sound
- The use of pre-trained methods in the mixture is straightforward and reproducible
- The results show improvements over the compared methods

**Weaknesses:**
- The design choices are not supported and are just stated
- The improvements of the claimed mixed local and global maps are not fully verified since the CLIP and diffusion methods are trained on different datasets.  Thus, the improvements could come from the different training sets rather than the local/global characteristics

**Audience:**

Yes

**Audience Explanation:**

Yes.  Anomaly detection is an interesting topic; and the use of open models with a straightforward incorporation of the features to be converted into anomaly scores may be of interest.

**Broader Impact Concerns:**

I have no concerns.

**Claims And Evidence:**

Yes

**Claims Explanation:**

Despite the weaknesses, the method claims to mix pre-trained models and to obtain better performance.  These claims are supported by the presented experiments according to the compared baselines.

The claim about the method being "training free" is misleading and should be reviewed.

**Requested Changes:**

- It is not clear what the authors mean by "entirely training-free" in the contributions.  The methods they are using are pre-trained, and the comments of what they are avoiding (fast deployment and sensitivity to training) still applies to the pre-trained models.  My guess is that they want to claim that using pre-trained models is beneficial, but it is a claim that should be revised, and even removed.

- Why use different encoders for the diffusion and the CLIP models?  Could the benefit come from the redundancy that the encoders provide rather than the different models (CLIP and diffusion)?  One way of evaluating this will be to have two CLIP/diffusion models with two different encoders trained independently and then do the mixture as well as the model-wise evaluation to verify where the gains are coming from.

- How are the abnormal conditions in Eq. 6 defined?  What is their impact in the final performance?

- Why use different strategies for the diffusion and CLIP scores? (Eqs. 10 vs. 11)  This selection should be supported empirically or theoretically.

- You should include pseudocode to fully define the behavior of your proposal.  While the parts are described throughout the paper, having a single point where the whole method is explained in detailed and in a clear way (with code), would help the reader to reproduce your work.

---

> ### Author Response · Authors · 2025-09-12
>
> We sincerely appreciate your valuable time and effort in reviewing our manuscript. We respond to each comment below.
>
> ---
> ### **[R1] Training-free method**
>
> The term "training-free" is widely used in the community when using foundation models as-is [1,2,3,4,5]. We also followed this practice, using the term "training-free" because we use CLIP and diffusion models for inference only, without additional training. However, to address the reviewer’s concern and improve clarity, the revised manuscript replaces this term with “no additional training.”
>
> [1] ComCLIP: Training-Free Compositional Image and Text Matching, NAACL'24.
>
> [2] FreeControl: Training-Free Spatial Control of Any Text-to-Image Diffusion Model with Any Condition, CVPR'24.
>
> [3] Pay Attention to Your Neighbours: Training-Free Open-Vocabulary Semantic Segmentation, WACV'25.
>
> [4] kNN-CLIP: Retrieval Enables Training-Free Segmentation on Continually Expanding Large Vocabularies, TMLR'24.
>
> [5] Explore the Potential of CLIP for Training-Free Open Vocabulary Semantic Segmentation, ECCV'24.
>
>
> ---
> ### **[R2] Different encoders for the diffusion and the CLIP models**
>
> CLIP and diffusion models contain their own encoder architecture. The encoder is not a separate entity but a component of each model. Our CLIP uses a vision transformer encoder, while our diffusion model uses a denoiser U-Net. Thus, different models already implies different encoders.
>
> ---
> ### **[R3] Abnormal conditions**
>
> The text prompts that describe abnormal conditions in Equation 6 are the prompts provided as textual input to the diffusion model (e.g., “a photo of a bottle with a crack”). We have described the prompts in the Prompts paragraph of Appendix A.2.
>
> In Appendix A.3, we report experiments on ensemble and customization of prompts. In addition, we conducted additional experiments using different prompt combinations. Below we report the results for the combinations. The results show our method remains reasonably robust across these variations. We have included the results in Appendix A.3.
>
> | States                                                | Task | MVTec-AD | VisA |
> |-------------------------------------------------------|------|----------|------|
> | `contamination`, `bend`, `cut`, `damage`              | Seg. | 95.8     | 97.0 |
> |                                                       | Cls. | 95.4     | 85.9 |
> | `break`, `fold`, `stain`, `damage`                    | Seg. | 95.8     | 97.3 |
> |                                                       | Cls. | 95.6     | 86.1 |
>
>
>
> ---
> ### **[R4] Strategies for the diffusion and CLIP scores**
>
> Equation 11 calculates the vision-guided scores of the CLIP and diffusion models, and it is the same strategy for both models because it takes the maximum from the score maps obtained from both models. This is a common practice adopted in several anomaly detection methods, including [1,2].
>
> Equation 10 is an equation for calculating the language-guided score in the diffusion model. We consider the peak-to-median ratio because we want to measure the degree of focus of the cross-attention map.
>
> [1] WinCLIP: Zero-/Few-Shot Anomaly Classification and Segmentation, CVPR'23.
>
> [2] ONE-FOR-ALL FEW-SHOT ANOMALY DETECTION VIA INSTANCE-INDUCED PROMPT LEARNING, ICLR'25.
>
>
> ---
> ### **[R5] Code to reproduce our work**
>
> Our code has already been submitted. Please see the attached .zip file in the supplementary materials.

---

> > ### Comment · Reviewer_akeK · 2025-09-12
> >
> > I thank the authors for their response.  Most of my concerns were resolved.
> >
> > Regarding the use common terms in the pre-trained models ("foundation" models) literature, I advise the authors to fix the terminology instead of embracing flawed terms.  It is only through usage that we will improve it.  So, I applaud their efforts to improve the final manuscript to be more clear.
> >
> > Regarding the different encoders, it weakens the results to simply use existing models but present them in a more general fashion.  From the description in the paper it seems that you have components that you can define, instead of reusing monolithic and static models.  It will be interesting to have the actual experiments where the components are trained and evaluated to fully understand their contribution.  I understand that this is an expensive tasks, but such is the burden given the selection of the models you are proposing.  If not, why not use a single encoder and use it as part in both the CLIP and diffusion formulation and train jointly instead?
> >
> > Regarding the code, I understand that having the open code is important and will fully supplement the manuscript.  However, for the longevity of your proposal, it is important to have self-contained ideas.  So, it will be beneficial to your proposal to have a pseudo-code that shows the overall behavior of your proposal.

---

> > > ### Author Response · Authors · 2025-09-15
> > >
> > > We thank the reviewer for the thoughtful suggestions.
> > >
> > > ---
> > > ### **[R1] Suggestions on joint training of a shared encoder for CLIP and diffusion models**
> > >
> > > Our work focuses on leveraging the complementarity of different foundation models (CLIP and diffusion) without any additional training. Effectively utilizing off-the-shelf foundation models is an active research direction in the current community. Our design aligns with this trend and is central to the practicality and novelty of our method.
> > >
> > > Using a single shared encoder and training it jointly for both CLIP and diffusion lies outside the scope and intent of our contribution. CLIP and diffusion have fundamentally different architectures and objectives (vision transformer vs. denoising U-Net, discriminative vs. generative). Forcing them into a shared-encoder formulation would not be straightforward and could compromise the unique strengths.
> > >
> > > We find the reviewer’s suggestion interesting but non-trivial, as it would require substantial additional ideas and experiments, and leave it as a potential avenue for future work.
> > >
> > > ---
> > > ### **[R2] Pseudocode**
> > >
> > > Thank you for the helpful suggestion. The overall idea and detailed step-by-step explanations of CLIPFUSION is provided in Section 4. To further improve clarity, we have added pseudocode in Appendix A.9. The pseudocode summarizes the workflow for anomaly segmentation and classification, and we indicate how each step corresponds to the equations and descriptions in the main text. We believe this addition makes the paper more accessible.

---

> > > > ### Comment · Reviewer_akeK · 2025-09-15
> > > >
> > > > I thank the authors for addressing my comments.
> > > >
> > > > Regarding the pseudocode, I understand that the explanations were in Section 4, but they were not enough.  I think that the added pseudocode in the Appendix will help to further clarify your explanations.
> > > >
> > > > Regarding the joint training, I understand that it is not trivial.  At least, I suggest that there is an explanation about your design choices and why you don't follow a joint encoder and rely on the existing architectures.

---

> > > > > ### Author Response · Authors · 2025-09-17
> > > > >
> > > > > ---
> > > > > ### **[R1] Joint training**
> > > > >
> > > > > We appreciate your suggestion. We have incorporated the rationale into the third paragraph of the Introduction.

---

### Review · Reviewer_GBSn · 2025-08-28

**Summary Of Contributions:**

The paper introduces CLIPFUSION, a training‑free framework that fuses a CLIP‑based branch (PatchCLIP) with a diffusion‑based branch (MapDiff) for anomaly segmentation and classification in both zero‑shot and few‑shot settings. On the diffusion branch, the author avoids doing iterative denoising; instead, they pass the clean image once through an inpainting pipeline with an empty mask to get cross‑attention maps and decoder feature maps, which contains rich spatial information for detecting anomalies. On the CLIP branch, they leverage patch‑level features  to provide a complementary global prior without sliding windows. The two branches are combined with fixed task‑dependent weights, yielding a single anomaly map or score. The method targets industrial benchmarks (MVTec‑AD, VisA), reports strong performance and speed, and also explores a “one‑for‑all” few‑shot setting in which a single model handles multiple object categories without any training.

**Audience:**

Yes

**Audience Explanation:**

Yes. The work integrates vision‑language modeling and diffusion‑based perception for anomaly detection, and it addresses a practical task of zero/few‑shot anomaly detection, where training data is scarce and deployment constraints matter. The idea of treating a text‑to‑image diffusion model as a feature extractor and fusing it with CLIP in a single, training‑free pipeline is likely to interest readers working on open‑vocabulary recognition and multimodal learning.

**Claims And Evidence:**

Yes

**Claims Explanation:**

Yes, across both datasets, CLIPFUSION outperforms prior methods in all four settings (zero/few‑shot × segmentation/classification). For example, in zero‑shot segmentation, AUROC improves from 85.1 to 90.9 on MVTec‑AD and from 79.6 to 92.1 on VisA over WinCLIP; in zero‑shot classification, it edges out WinCLIP on both datasets as well. Tables 1–2 document these margins with averages across five seeds and standard deviations.

The author also implements ablation studies for the contribution of each model and modality. Using only CLIP or only diffusion degrades performance; combining them obtains the best performance for both tasks (Table 3). Splitting language‑guided and vision‑guided pathways shows that each brings complementary information (Table 4). These ablations directly substantiate the core premise that discriminative and generative foundations capture different, useful signals.

**Requested Changes:**

1 A strength of the paper is extensive per‑class reporting (Tables 14–21, pages 19–20). These tables also show a few informative counter‑examples. On MVTec zero‑shot classification, CLIPFUSION trails WinCLIP on Screw and Transistor (Table 14). On MVTec zero‑shot segmentation, Cable is lower than WinCLIP (Table 16). On VisA zero‑shot classification, several categories (e.g., Candle, Cashew, PCB4) are slightly behind WinCLIP (Table 18). A short failure analysis that overlays PatchCLIP and MapDiff maps for these cases, e.g., are diffusion attentions saturating on repetitive textures, or is CLIP over‑focusing on global shape? This would make the paper more convincing. It would also tell readers when to raise or lower the diffusion weight in practice.

2 All results use Stable Diffusion v2 as the denoiser, with images resized to 512 for MapDiff and to 240 for OpenCLIP (Appendix A.2). Since the core claim is that “diffusion as a feature extractor” is broadly useful, it would strengthen the paper to show backbone sensitivity (e.g., a second diffusion checkpoint with different text encoders or training data) and prompt sensitivity beyond the state‑word ensemble (Appendix A.2–A.3).

3 The proposed framework relies on a small, manually curated set of “state” words (damage, crack, hole, residue), with optional per-class customization, and then averages the resulting cross‑attention maps (Figure 3, page 6; Appendix A.2–A.3). This works well when the anomaly matches the list, but may not work well with defects that are subtle, composite, or product‑specific.

---

> ### Author Response · Authors · 2025-09-12
>
> We sincerely appreciate your valuable time and effort in reviewing our manuscript. We respond to each comment below.
>
>
>
> ---
> ### **[R1] Failure case analysis**
>
> We qualitatively analyzed the anomaly maps and identified two representative failure cases, which are included in Appendix A.7. One of them corresponds to the reviewer’s hypothesis: we observed attention saturation, where the diffusion model tends to over-activate on dense and repetitive patterns. We sincerely thank the reviewer for this suggestion, which helped us strengthen the paper.
>
> ---
> ### **[R2] Different backbones and prompts**
>
> We evaluated our framework on multiple diffusion backbones. The results below remain highly consistent across backbones, with only marginal differences. This shows that our method is robust to the choice of diffusion backbone. We have included the results in Appendix A.8.
>
> | Backbone                    | Task | MVTec-AD | VisA |
> | ------------------------- | ---- | -------- | ---- |
> | **Stable Diffusion v1.5** | Seg. | 95.8     | 96.7 |
> |                           | Cls. | 93.5     | 86.4 |
> | **Stable Diffusion v2.0** | Seg. | 95.8     | 97.3 |
> |                           | Cls. | 95.4     | 86.5 |
> | **Stable Diffusion v2.1** | Seg. | 95.8     | 97.3 |
> |                           | Cls. | 94.6     | 86.5 |
>
>
> We conducted additional experiments using different prompts. Below we report the results for the combinations. The results show our method remains reasonably robust across these variations. We have included the results in Appendix A.3.
>
> | States                                                | Task | MVTec-AD | VisA |
> |-------------------------------------------------------|------|----------|------|
> | `contamination`, `bend`, `cut`, `damage`              | Seg. | 95.8     | 97.0 |
> |                                                       | Cls. | 95.4     | 85.9 |
> | `break`, `fold`, `stain`, `damage`                    | Seg. | 95.8     | 97.3 |
> |                                                       | Cls. | 95.6     | 86.1 |
>
> ---
> ### **[R3] On the reliance on a set of state words**
>
> When practitioners apply this technology to a specific product in a real-world scenario, they usually have some knowledge of the anomaly types that can occur in that product. To cover anomaly types that fall outside this prior domain knowledge, we include a generic state word like "damage."

---

### Review · Reviewer_VVJ4 · 2025-09-06

**Summary Of Contributions:**

This paper introduces a novel CLIPFUSION framework for anomaly detection based on CLIP and diffusion models. The proposed framework allows to extract intermediate features from an off-the-shelf diffusion model without any training or iterative denoising. Extensive experimental results demonstrate the effectiveness and computational efficiency of the CLIPFUSION framework in multiple benchmarks.

**Audience:**

Yes

**Audience Explanation:**

This work provided a simple yet effective anomaly detection approach. Thus, TMLR's audience will be interested in the proposed framework as well as the empirical results shown in the experiments.

**Broader Impact Concerns:**

No explicit ethical concerns in this work.

**Claims And Evidence:**

Yes

**Claims Explanation:**

The motivation of the proposed framework for integrating CLIP and diffusion models is clearly explained. The technical details are easy to follow.

**Requested Changes:**

(1) The rationale behind the proposed framework is not well-explained. For example, it is unclear why the cross-attention mechanism and each of the score maps in Section 4.1 can improve anomaly detection.

(2) Following previous concerns, one key hyperparameter in the proposed framework is the weight $\alpha$ between the models. As shown in experimental settings, classification and segmentation require different values of $\alpha$. However, the impact of this hyperparameter is not discussed. Besides, it is unclear how the value of $\alpha$ is selected for the one-for-all paradigm results in Table 8.

---

> ### Author Response · Authors · 2025-09-12
>
> We sincerely appreciate your valuable time and effort in reviewing our manuscript. We respond to each comment below.
>
> ---
> ### **[R1] Rationale behind the proposed framework**
>
> The cross-attention map of a diffusion model is a heatmap showing where and how strongly a text token is attended to across image locations. Because we use a token that denotes an anomalous state (e.g., ‘damage’), the attention weight at a location serves as a proxy for anomaly likelihood.
>
> Regarding feature maps, we extract normal features from normal reference images by passing them through the CLIP image encoder and the diffusion denoiser. Then, the query image is passed through and the extracted features are compared with the normal features. The degree of deviation is used as an anomaly score.
>
> Section 4.1 originally provided an explanation of the rationale. We have now further supplemented this section by adding intuition for each score map.
>
> ---
> ### **[R2] Weights between the two models**
>
> In the revised version, we report in Appendix A.6 the performance under different alpha values. $\alpha$ and $(1-\alpha)$ denote the weights of the CLIP and the diffusion model, respectively. By changing the weights between the two models, we adjust their relative contributions and evaluate the performance. As observed in the following experiments, segmentation achieves the best performance among the tested alpha values at $\alpha = 0.25$, while classification performed best at $\alpha = 0.75$.
>
> | Task | CLIP | Diff | MVTec-AD | VisA |
> |------|------|------|----------|------|
> | **Seg** |      |      |          |      |
> |        | 1    | 0    | 94.7     | 93.7 |
> |        | 0.75 | 0.25 | 95.4     | 96.1 |
> |        | 0.5  | 0.5  | 95.7     | 97.2 |
> |        | 0.25 | 0.75 | **95.8** | **97.3** |
> |        | 0    | 1    | 95.1     | 96.5 |
> | **Cls** |      |      |          |      |
> |        | 1    | 0    | 93.5     | 82.2 |
> |        | 0.75 | 0.25 | **95.4** | **86.5** |
> |        | 0.5  | 0.5  | 95.0     | 86.4 |
> |        | 0.25 | 0.75 | 92.7     | 83.9 |
> |        | 0    | 1    | 87.2     | 78.9 |
>
> For the one-for-all paradigm in Table 8, we used the same experimental setting as in Section 5, including the choice of alpha. We have clarified this point in the revised manuscript.

---

### Public Comment · ~Tai_Le_Gia1 · 2025-09-12
**Questions about the paper**

Dear Authors,

Thank you for your interesting submission on CLIPFUSION for anomaly detection. I have a few questions and clarifications I'd like to raise for better understanding. I'd appreciate any insights you can provide.

1. Regarding the alignment in PatchCLIP for the zero-shot language-guided score map (Section 4.1.1): You mention calculating the degree (Equation 1) for each patch embedding's alignment to the abnormal text embedding. However, Equation 1 is originally defined for the global image embedding (typically the CLS token) versus text. Could you clarify how this is adapted for individual patch embeddings? For instance, are the patch embeddings projected to the multimodal space (e.g., from 768 to 512 dimensions in ViT-B/16) to ensure compatibility with text embeddings, or is there another adjustment? For examples, in APRIL-GAN [2], they introduced a projection to align patch embeddings and text embeddings. In AnomalyCLIP [3], they fine-tuned text prompts against the patch embeddings after projecting them using CLIP's ln_post.

2. On the complementarity between the CLIP encoder (discriminative) and diffusion denoiser (generative): The paper highlights how diffusion compensates for CLIP's limitations in fine-grained local details. Could you elaborate on the underlying reasons for this complementarity? Is it primarily due to differences in training datasets (as reviewer akeK asked), or more from the architectures/training methods (e.g., direct embedding alignment in CLIP vs. indirect generative alignment in diffusion)? For comparison, in [1] (Asymmetric Student-Teacher Networks for Industrial Anomaly Detection), asymmetric architectures are used to avoid undesired generalization in teacher-student pairs—does a similar principle (difference in architecture of two encoder) apply here?

3. In Appendix A.6 (Effect of weights between the CLIP and diffusion models), could you specify which setting (zero-shot or k-shot) the reported results correspond to?

4. In the related work (Section 2), you note that "MuSc (Li et al., 2024b) leverages the test set by exhaustively comparing the features of all test images, which is only feasible when the entire test set is available and contains a low proportion of anomalies." However, the MuSc implementation includes a parameter (--divide_num) to divide the test set into subsets, allowing processing of smaller chunks (i.e., requiring only a portion of the test dataset). Does this affect the characterization, or am I missing something?

Thanks again for your work—looking forward to any responses!

Best regards,

References:
[1] Rudolph et al., Asymmetric Student-Teacher Networks for Industrial Anomaly Detection, WACV 2023.

[2] Chen et al., APRIL-GAN: A Zero-/Few-Shot Anomaly Classification and Segmentation Method for CVPR 2023 VAND Challenge Track 1 AWARD, arXiv:2305.17382.

[3] Hang et al., AnomalyCLIP: Object-agnostic Prompt Learning for Zero-shot Anomaly Detection, ICLR 2024.

---

> ### Author Response · Authors · 2025-09-15
>
> Thank you for your interest in our work. We address each point below.
>
> ---
> ### **[Q1] Alignment in PatchCLIP**
>
> As stated in Section 4.1.1, we directly compute the similarity between each patch embedding (obtained together with the class token embedding) and the text embedding. Please refer to the `get_clip_language_score_map` function in `core.py` in our released code.
>
> ```
> _, token_features, _, _ = model.encode_image(image)
> token_features = token_features.squeeze(0)
> token_features = token_features / token_features.norm(dim=-1, keepdim=True)
>
> token_scores = (token_features @ text_features.T).softmax(dim=-1)[:, 1]
> ```
>
> ---
> ### **[Q2] Complementarity between the CLIP encoder and diffusion denoiser**
>
> It is known that CLIP is limited in capturing fine-grained local details, as its discriminative training enforces global alignment between image and text features. In contrast, diffusion models can capture such fine-grained details, since their denoising objective requires learning local semantics to generate images from noise. In this work, we find that diffusion features complement CLIP features qualitatively and quantitatively.
>
> ---
> ### **[Q3] Setting in Appendix A.6**
>
> As in Section 6, the results are based on the one-shot setup.
>
> ---
> ### **[Q4] On the MuSc evaluation setting**
>
> The main results reported in the MuSc paper correspond to using the full test set at once. Since this requires substantial memory and time, the `--divide_num` option is introduced as an engineering choice to split the test set into smaller chunks for efficiency. As reported in the paper, this leads to a drop in performance. In this case the entire test set is still used, only processed sequentially in subsets.

---

> ### Public Comment · ~Tai_Le_Gia1 · 2025-09-17
> **Questions about the paper**
>
> Dear Authors,
>
> Thank you for your detailed responses and the provided code snippet. I have follow-up questions for Q1 and a suggested revision for Q4 to ensure clarity and alignment with the referenced works.
>
> Q1: Alignment in PatchCLIP for Zero-Shot Language-Guided Score Map
>
> The WinCLIP paper (Sections 4.2 and 5.4) highlights that directly using last-layer patch embeddings results in poor dense predictions (e.g., Table 8: pAUROC of 22.4% and PRO of 2.3% for the "Patch-token" baseline) due to insufficient alignment with the language space. This necessitates additional fine-tuning in methods like proposed in AnomalyCLIP [3] and APRIL-GAN [2]. Your response suggests a "direct" similarity computation between patch embeddings (token_features) and text embeddings. To clarify how your approach overcomes WinCLIP's reported limitations, could you address the following:
>
>
>
>
>
> - How do you project patch embeddings to the same space as text embeddings? For example, in CLIP ViT-B/16, patch embeddings are 768-dimensional, while text embeddings are 512-dimensional—do you use CLIP's visual projection layer (e.g., visual.ln_post followed by projection) to align them?
>
>
>
> - If such a projection is applied, why does your direct use of patch embeddings without fine-tuning on auxiliary datasets yield viable performance, unlike WinCLIP's poor results for unadapted patches (Table 8 section 5.4)?
>
> Q4: Characterization of MuSc in Related Work (Section 2)
>
> MuSc's --divide_num option  allows processing the test set in independent chunks. Hence I suggest revising the statement to: "MuSc is only feasible for batch-processing/batch-zero-shot" rather than "only feasible when the entire test set is available," to reflect this capability, as suggested in [5].
>
> Thank you again for your insightful work—I look forward to your clarifications.
>
> Best regards,
>
> Reference:
>
> [4] Jeong, Jongheon, et al. "Winclip: Zero-/few-shot anomaly classification and segmentation." Proceedings of the IEEE/CVF Conference on Computer Vision and Pattern Recognition. 2023.
> [5] Damm, Simon, et al. "Anomalydino: Boosting patch-based few-shot anomaly detection with dinov2." 2025 IEEE/CVF Winter Conference on Applications of Computer Vision (WACV). IEEE, 2025.

---

> > ### Author Response · Authors · 2025-09-17
> >
> > ---
> > ### **[Q1] Alignment in PatchCLIP for Zero-Shot Language-Guided Score Map**
> >
> > We use OpenCLIP ViT-B-16-plus-240, whose shared CLIP embedding dimension is 640. We use the model as is and did not add extra projector. This can be confirmed in our released code. In our experiments, we observed that patch embeddings yield reasonable performance, and we report these empirical results as observed. These results can be directly verified by running our released code. Since WinCLIP has not publicly released the code for this baseline, it is difficult to reproduce or analyze their exact implementation for this case, and we therefore refrain from speculating.
> >
> > ---
> > ### **[Q2] Characterization of MuSc in Related Work (Section 2)**
> >
> > We have revised the statement to reflect your suggestion.

---

> ### Public Comment · ~Tai_Le_Gia1 · 2025-09-19
> **Questions about the paper**
>
> Dear Authors,
>
> Thank you for the additional details on your model choice and for revising the MuSc statement—much appreciated.
>
> Regarding Q1: You use OpenCLIP’s ViT-B-16-plus-240 without extra projectors for patch embedding alignment. However, the model configuration https://github.com/mlfoundations/open_clip/blob/main/src/open_clip/model_configs/ViT-B-16-plus-240.json shows a dimensional mismatch: the vision transformer’s internal dimension is 896, while the shared multimodal embedding space (including text) is 640. This raises a natural question about how direct similarity computation between patch and text embeddings is enabled. (Note: I could not locate your released source code in the provided zip file—possibly due to limited access as a viewer rather than a reviewer—so I am relying on this config to infer the setup.) To clarify how your approach overcomes this, could you address the following:
>
> 1. How do you handle the dimensional mismatch between the 896-dim visual patch embeddings and the 640-dim text embeddings in your implementation?
>
> 2. Note that WinCLIP reports significantly lower performance when using unadapted patch embeddings with ViT-B-16-plus-240 (e.g., pAUROC 22.4%, PRO 2.3% in Table 8 for "Patch-token"), despite using the same base model. Given this, it would be valuable to discuss why your approach achieves strong results without auxiliary fine-tuning—particularly in light of WinCLIP’s findings. Could this be due to your implementation choices that help mitigate embedding misalignment? and how?  I think this would significantly strengthen the paper.
>
> I look forward to your insights.
>
> Best regards,

---

> > ### Author Response · Authors · 2025-09-20
> >
> > The vision transformer internally uses 896 dimensions, but the final output dimension is aligned to the shared embedding space (640) through the model’s built-in projection. Please refer to the following link for implementation details:
> >
> > https://github.com/mlfoundations/open_clip/blob/main/src/open_clip/model.py#L194C13-L194C34
> >
> > On the second question, our position has been stated in the earlier response, and we kindly refer you to that explanation.

---

### Decision · Action_Editor_nThQ · 2025-10-14

**Recommendation:** Accept as is

**Audience:**

Yes

**Audience Explanation:**

The paper addresses the practical and relevant problem of zero/few-shot anomaly detection with recent techniques such as CLIP and diffusion models, which will be of interest to many in the field of anomaly detection and beyond.

**Claims And Evidence:**

Yes

**Claims Explanation:**

The paper leverages a combination of pre-trained vision-language models (CLIP) and diffusion models for few-shot and zero-shot anomaly detection. The method is demonstrated to outperform baselines on standard benchmark tasks, and ablation studies show that indeed the CLIP and diffusion models are complementary.

There were initial common concerns that design choices were not well explained but this was largely addressed in the revision to the satisfaction of the reviewers. All reviewers unanimously agreed that post-revision, the paper's claims are well supported.